# Flow Matching with Uncertainty Quantification and Guidance

## Abstract

Despite the remarkable success of sampling-based generative models such as flow matching, they can still produce samples of inconsistent or degraded quality. To assess sample reliability and generate higher-quality outputs, we propose *uncertainty-aware flow matching* (UA-Flow), a lightweight extension of flow matching that predicts the velocity field together with heteroscedastic uncertainty. UA-Flow estimates per-sample uncertainty by propagating velocity uncertainty through the flow dynamics. These uncertainty estimates act as a reliability signal for individual samples, and we further use them to steer generation via uncertainty-aware classifier guidance and classifier-free guidance. Experiments on image generation show that UA-Flow produces uncertainty signals more highly correlated with sample fidelity than baseline methods, and that uncertainty-guided sampling further improves generation quality.

## 1. Introduction

In recent years, sampling-based generative models such as diffusion (Sohl-Dickstein et al., 2015; Ho et al., 2020; Song et al., 2021) and flow matching (Lipman et al., 2023; Liu et al., 2023; Albergo & Vanden-Eijnden, 2023) have achieved remarkable success across a wide range of domains, most particularly in image and video generation (Dhariwal & Nichol, 2021; Ho et al., 2022) as well as in sequential decision-making (Janner et al., 2022; Chi et al., 2025; Black et al., 2024). Despite this progress, these models often produce samples of inconsistent quality. As a result, using reliably generated samples in downstream applications remains challenging, and this challenge becomes particularly critical in safety-sensitive settings such as robotics and decision-making.

To address this issue, the uncertainty associated with each generated sample can be interpreted as a measure of the reliability of the generation process. Recently, several works have explored uncertainty estimation for diffusion-based generative models by adapting techniques from existing uncertainty quantification literature for neural networks (Lakshminarayanan et al., 2017; Kendall & Gal, 2017; Ritter et al., 2018). In sampling-based generative modeling, uncertainty plays two central roles: (i) it provides a principled signal for assessing the quality or reliability of individual generated samples, and (ii) uncertainty can be leveraged during the generative process to actively improve sample quality through guided sampling (De Vita & Belagiannis, 2025). However, most prior works (Kou et al., 2023; Jazbec et al., 2025) primarily focus on the first role, using uncertainty in a post-hoc manner for sample filtering or selection. Moreover, existing approaches are often domain-specific (Sun et al., 2023; Franchi et al., 2025) and quantify uncertainty over conditional inputs (Berry et al., 2024).

In this work, we propose *uncertainty-aware flow matching* (UA-Flow), a lightweight extension of flow matching that models heteroscedastic uncertainty in the velocity field. By propagating this velocity uncertainty through the flow dynamics, UA-Flow provides principled sample uncertainty estimates with minimal additional overhead. Because uncertainty is modeled at the velocity level, our approach does not rely on domain-specific uncertainty representations. Moreover, we can leverage the learned velocity uncertainty for uncertainty-aware guided sampling, which improves generation quality and is not explicitly considered in closely related prior work (Kou et al., 2023; Jazbec et al., 2025). The deterministic sampling allows us to localize uncertainty to the learned velocity field and propagate it through the dynamics, in contrast to stochastic sampling.

We empirically validate uncertainty estimation with our proposed approach in two settings. First, we provide evidence that UA-Flow's uncertainty correlates with sample fidelity, with higher uncertainty indicating lower fidelity. In particular, filtering out high-uncertainty samples yields better fidelity-oriented metrics than prior uncertainty-quantification baselines for sampling-based generative models (Kou et al., 2023; De Vita & Belagiannis, 2025). Second, our systematic ablation studies demonstrate that uncertainty reduction can be actively incorporated as guidance

[1]Anonymous Institution, Anonymous City, Anonymous Region, Anonymous Country. Correspondence to: Anonymous Author <anon.email@domain.com>.

Preliminary work. Under review by the International Conference on Machine Learning (ICML). Do not distribute.

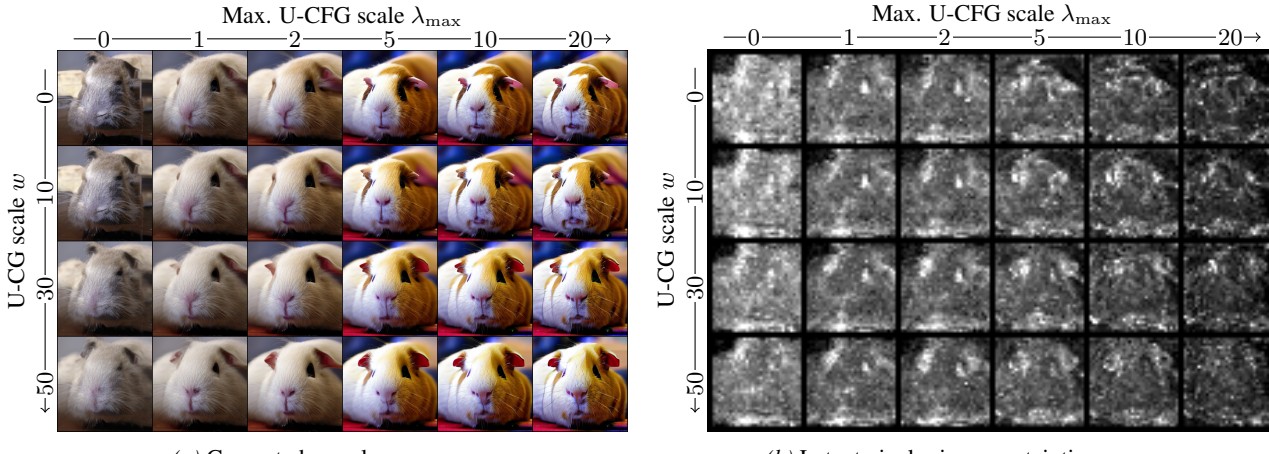

*(a)* Generated samples.

*(b)* Latent pixel-wise uncertainties.

*Figure 1.* **Uncertainty-aware guidance sweep on ImageNet-256: generated samples and predicted latent pixel-wise uncertainties under uncertainty-aware classifier and classifier free guidance (U-CG and U-CFG).** We visualize ImageNet-256 samples for the class `guinea pig, Cavia cobaya` across combinations of U-CG and U-CFG. Rows sweep the U-CG scale $w \in \{0, 10, 30, 50\}$ and columns sweep the maximum U-CFG scale $\lambda_{\max} \in \{0, 1, 2, 5, 10, 20\}$. The left panel shows generated images, and the right panel shows the corresponding predicted uncertainty maps (brighter indicates higher uncertainty). Increasing guidance typically yields more class-consistent samples while reducing predicted uncertainty.

during sampling, where both uncertainty-aware classifier and classifier-free guidance lead to improved generation quality.

## 2. Background

**Sampling-based generative models and guidance.** Sampling-based generative models synthesize data by iteratively transforming samples from a simple base distribution into the data distribution. Diffusion models implement this transformation through a gradual denoising process (Sohl-Dickstein et al., 2015; Ho et al., 2020; Song et al., 2021). Flow matching instead learns a deterministic ordinary differential equation (ODE) whose velocity field transports samples along a prescribed probability path (Lipman et al., 2023; Liu et al., 2023; Albergo & Vanden-Eijnden, 2023). Across both families, generation can be substantially improved by guidance, which modifies the sampling dynamics to favor samples that better satisfy a condition. Classifier guidance (CG) injects gradients from an external classifier or constraint functions into the sampling update (Dhariwal & Nichol, 2021; Dao et al., 2023), while classifier-free guidance (CFG) extrapolates conditional and unconditional predictions without a separate classifier (Ho & Salimans, 2022; Zheng et al., 2023). A practical challenge is that strong guidance can reduce diversity or induce artifacts at high scales, motivating approaches that moderate the effective guidance signal or adapt its strength during sampling (Saharia et al., 2022; Sadat et al., 2024).

**Uncertainty quantification for neural networks.** Uncer-

tainty quantification (UQ) for neural networks assesses prediction reliability and is commonly decomposed into aleatoric uncertainty (data-dependent noise) and epistemic uncertainty (model uncertainty) (Kendall & Gal, 2017). A widely used approach to aleatoric UQ is heteroscedastic regression, where the network predicts both a mean and an input-dependent variance and is trained with a Gaussian negative log-likelihood (Nix & Weigend, 1994). Epistemic uncertainty is often approximated with Bayesian-inspired techniques such as Monte Carlo dropout, deep ensembles, or Laplace approximations (Gal & Ghahramani, 2016; Lakshminarayanan et al., 2017; Ritter et al., 2018). While these methods are well-established in classification and regression, applying them to sampling-based generative models requires specifying *which* intermediate quantities are uncertain (e.g., score/velocity or conditional input) and *how* uncertainty propagates through the sampling dynamics.

**Uncertainty quantification with sampling-based generative models.** Most existing UQ methods for sampling-based generative models are diffusion-centric and can be broadly grouped by *where* uncertainty is modeled. Some quantify uncertainty in the conditional input of conditional diffusion (Berry et al., 2024). Others develop diffusion-model uncertainty for domain-specific generation and decision-making, including trajectory planning and multi-agent forecasting (Sun et al., 2023; Capellera et al., 2025), as well as text-to-image uncertainty analysis (Franchi et al., 2025). A third line targets uncertainty of the generated sample itself. BayesDiff (Kou et al., 2023) uses Bayesian/Laplace-based estimators (Daxberger et al., 2021) and propagates uncertainty through the diffusion dynamics. Other meth-

ods rely on feature-space likelihoods based on CLIP encoders (Radford et al., 2021; Jazbec et al., 2025), or pixel-wise aleatoric uncertainty for reliability scoring and uncertainty-guided sampling (De Vita & Belagiannis, 2025). Despite this progress, flow-matching-specific UQ remains underexplored, even though its deterministic ODE structure enables modeling uncertainty in the learned velocity field and propagating it through the dynamics.

## 3. Uncertainty-Aware Flow Matching

This section presents *uncertainty-aware flow matching* (UA-Flow), which learns heteroscedastic uncertainty directly in the velocity field and propagates it through the generative flow. The resulting uncertainty enables uncertainty-aware classifier and classifier-free guidance during sampling.

**Setup and notation.** Flow matching learns a time-dependent velocity field that transports samples from a base distribution $\mathbf{x}_0 \sim p_0$ to data $\mathbf{x}_1 \sim p_1$ along a prescribed probability path. We adopt the common affine path $\mathbf{x}_t = \alpha_t \mathbf{x}_1 + \beta_t \mathbf{x}_0$ over time $t \in [0, 1]$, which induces the conditional distribution $p_t(\mathbf{x}_t \mid \mathbf{x}_1)$ and the corresponding closed-form conditional target velocity $u_t(\mathbf{x}_t \mid \mathbf{x}_1)$ (Lipman et al., 2024).

### 3.1. Probabilistic Velocity Field Modeling

UA-Flow aims to learn both the mean, $\bar{u}_t^\theta(\mathbf{x}_t) \in \mathbb{R}^n$, and the diagonal variance, $(\sigma_t^\theta(\mathbf{x}_t))^2 \in \mathbb{R}^n$, of the velocity field, using the target velocity $u_t(\mathbf{x}_t)$ as supervision. For the computational efficiency and representational simplicity, we estimate the variance in an element-wise manner. The uncertainty-aware flow matching loss, $\mathcal{L}_{\mathrm{UFM}}(\theta)$, is formulated as Gaussian negative log-likelihood (NLL) loss targeting the velocity:

$$\mathcal{L}_{\mathrm{UFM}}(\theta) = \mathbb{E}_{t,p_t(\mathbf{x}_t)} \left[ \frac{\left(\bar{u}_t^\theta(\mathbf{x}_t) - u_t(\mathbf{x}_t)\right)^2}{2(\sigma_t^\theta(\mathbf{x}_t))^2} + \log(\sigma_t^\theta(\mathbf{x}_t)) \right]. \quad (1)$$

All operations in $\mathcal{L}_{\mathrm{UFM}}$ are applied element-wise. This convention is used throughout the paper.

As in standard flow matching, the target velocity $u_t(\mathbf{x}_t)$ is not directly accessible, and training instead regresses the model to the conditional velocity $u_t(\mathbf{x}_t \mid \mathbf{x}_1)$ under $p_t(\mathbf{x}_t \mid \mathbf{x}_1)$. Following the same principle, UA-Flow minimizes a conditional uncertainty-aware flow matching loss, denoted by $\mathcal{L}_{\mathrm{CUFM}}$, which reformulates $\mathcal{L}_{\mathrm{UFM}}$ in terms of the conditional velocity:

$$\mathcal{L}_{\mathrm{CUFM}}(\theta) = \mathbb{E}_{t,p_1(\mathbf{x}_1),p_t(\mathbf{x}_t \mid \mathbf{x}_1)} \left[ \frac{U_t(\mathbf{x}_t, \mathbf{x}_1)}{2(\sigma_t^\theta(\mathbf{x}_t))^2} + \right.$$
$$\left. \frac{\left(\bar{u}_t^\theta(\mathbf{x}_t) - u_t(\mathbf{x}_t \mid \mathbf{x}_1)\right)^2}{2(\sigma_t^\theta(\mathbf{x}_t))^2} + \log(\sigma_t^\theta(\mathbf{x}_t)) \right], \quad (2)$$

where $U_t(\mathbf{x}_t, \mathbf{x}_1) := \hat{u}_t(\mathbf{x}_t)^2 - u_t(\mathbf{x}_t \mid \mathbf{x}_1)^2$ is a correction term between the unconditional velocity estimate, $\hat{u}_t(\mathbf{x})$, and the conditional velocity. In practice, we define the estimated target velocity $\hat{u}_t(\mathbf{x})$ using the reweighted mini-batch estimator:

$$\hat{u}_t(\mathbf{x}_t) = \frac{\sum_{b=1}^B u_t(\mathbf{x}_t \mid \mathbf{x}_{1,b}) p_t(\mathbf{x}_t \mid \mathbf{x}_{1,b})}{\sum_{b=1}^B p_t(\mathbf{x}_t \mid \mathbf{x}_{1,b})}. \quad (3)$$

We detail the approximation from $\mathcal{L}_{\mathrm{UFM}}(\theta)$ to $\mathcal{L}_{\mathrm{CUFM}}(\theta)$ and analyze $U_t(\mathbf{x}_t, \mathbf{x}_1)$ in Appendix A.

**Remark.** UA-Flow can be fine-tuned from a pre-trained flow matching model, without requiring training from scratch. In practice, we find that initializing from a pre-trained model is beneficial for preserving generation quality while learning uncertainty. Moreover, we adopt the $\beta$-NLL loss (Seitzer et al., 2022) rather than the standard Gaussian NLL, which scales each sample loss by $\mathrm{sg}\left[(\sigma_t^\theta(\mathbf{x}_t))^{2\beta}\right]$ with $\beta \in [0, 1]$ because it produces better mean estimates.

### 3.2. Uncertainty Propagation through Flow Dynamics

We aim to estimate uncertainty of a generated sample reflecting accumulated velocity uncertainty along the flow. Specifically, given the predicted mean and variance of $u_t^\theta(\mathbf{x}_t)$, $\bar{u}_t^\theta(\mathbf{x}_t)$ and $(\sigma_t^\theta(\mathbf{x}_t))^2$, we propagate the mean $\bar{\mathbf{x}}_t$ and variance $\mathrm{Var}[\mathbf{x}_t]$ of the state $\mathbf{x}_t$ starting from the initial state $\mathbf{x}_0$ sampled from the base distribution $p_0$. We interpret the resulting mean and variance at the final time, $\bar{\mathbf{x}}_1$ and $\mathrm{Var}[\mathbf{x}_1]$, as the generated sample and its associated uncertainty.

To obtain the mean dynamics, we linearize $\bar{u}_t^\theta(\mathbf{x}_t)$ around $\bar{\mathbf{x}}_t$ and drop higher-order terms, yielding

$$\frac{d\bar{\mathbf{x}}_t}{dt} = \bar{u}_t^\theta(\bar{\mathbf{x}}_t). \quad (4)$$

For variance propagation we adopt Euler discretization, $\mathbf{x}_{t+\Delta t} = \mathbf{x}_t + u_t^\theta(\mathbf{x}_t)\Delta t$, since variance propagation using higher-order solvers would require a substantial increase in analytical complexity and computational cost. Similar to the variance propagation of BayesDiff (Kou et al., 2023), we approximate the evolution of the element-wise variance from $\mathbf{x}_t$ to $\mathbf{x}_{t+\Delta t}$ as

$$\mathrm{Var}[\mathbf{x}_{t+\Delta t}] \approx \mathrm{Var}[\mathbf{x}_t] + (\sigma_t^\theta(\bar{\mathbf{x}}_t)\Delta t)^2 + 2\Delta t \mathrm{Cov}(\mathbf{x}_t, u_t^\theta(\mathbf{x}_t)), \quad (5)$$

Here, $(\sigma_t^\theta(\bar{\mathbf{x}}_t))^2$ is the predicted velocity variance evaluated at the mean state, and $\mathrm{Cov}(\mathbf{x}_t, u_t^\theta(\mathbf{x}_t)) \in \mathbb{R}^n$ denotes the element-wise covariance between the state and its velocity. The former quantifies injected velocity noise, while the latter captures how state uncertainty couples with the local sensitivity of the velocity field.

Applying a first-order Taylor expansion again, the element-wise covariance $\mathrm{Cov}(\mathbf{x}_t, u_t^\theta(\mathbf{x}_t))$ is approximated as

$$\mathrm{Cov}(\mathbf{x}_t, u_t^\theta(\mathbf{x}_t)) \approx \mathrm{diag}(J_t^\theta(\bar{\mathbf{x}}_t)) \odot \mathrm{Var}[\mathbf{x}_t]. \quad (6)$$

where $J_t^\theta(\bar{\mathbf{x}}_t) = \left.\frac{\partial \bar{u}_t^\theta}{\partial \mathbf{x}_t}\right|_{\bar{\mathbf{x}}_t} \in \mathbb{R}^{n \times n}$ denotes the Jacobian of the mean velocity with respect to the state, evaluated at $\bar{\mathbf{x}}_t$. Also, $\odot$ represents element-wise multiplication. Since forming $\mathrm{diag}(J_t^\theta(\bar{\mathbf{x}}_t))$ explicitly is intractable in high dimensions, we approximate $\mathrm{diag}(J_t^\theta(\bar{\mathbf{x}}_t)) \odot \mathrm{Var}[\mathbf{x}_t]$ using Hutchinson's diagonal estimator (Bekas et al., 2007; Dharangutte & Musco, 2023). Consequently, the covariance $\mathrm{Cov}(\mathbf{x}_t, u_t^\theta)$ can be estimated as:

$$\mathrm{Cov}(\mathbf{x}_t, u_t^\theta(\mathbf{x}_t)) \approx \frac{1}{S} \sum_{i=1}^{S} (\boldsymbol{\sigma}_t^x \odot \mathbf{r}_i) \odot (J_t^\theta(\bar{\mathbf{x}}_t)(\boldsymbol{\sigma}_t^x \odot \mathbf{r}_i)) \quad (7)$$

with $\boldsymbol{\sigma}_t^x = \sqrt{\mathrm{Var}[\mathbf{x}_t]}$. $\mathbf{r}_i \in \mathbb{R}^n$ is a Rademacher vector whose entries are independently sampled from $\{-1, +1\}$ with equal probability. Using Jacobian-vector products (JVPs) (Baydin et al., 2018), $J_t^\theta(\bar{\mathbf{x}}_t)(\boldsymbol{\sigma}_t^x \odot \mathbf{r}_i)$ can be computed efficiently via automatic differentiation without forming $J_t^\theta$ explicitly. Algorithm 1 summarizes the Monte Carlo estimator corresponding to Equation (7).

We provide full derivations of Equations (4) to (7), as well as alternative covariance approximations, in Appendix B.

### 3.3. Uncertainty-Aware Guidance for Flow Matching

We incorporate the predicted velocity uncertainty into guided sampling by modifying the mean dynamics in Equation (4). We present two mechanisms: (i) an uncertainty-based pseudo-likelihood whose gradient is used as a classifier-guidance term, and (ii) an adaptive choice of the CFG scale that reduces the predicted variance of the extrapolated velocity.

**Uncertainty-aware classifier guidance (U-CG).** Standard classifier guidance modifies the sampling dynamics by adding a term $b_t w \nabla_\mathbf{x} \log p_t(y \mid \mathbf{x})$ to the velocity field, where $w \geq 0$ is the guidance scale. For the affine path $\mathbf{x}_t = \alpha_t \mathbf{x}_1 + \beta_t \mathbf{x}_0$, $b_t = -\frac{\dot{\beta}_t \beta_t \alpha_t - \dot{\alpha}_t \beta_t^2}{\alpha_t}$. We define an uncertainty-based pseudo-likelihood over states, $\tilde{p}_t(\bar{\mathbf{x}}_t) \propto \exp(f((\sigma_t^\theta(\bar{\mathbf{x}}_t))^2))$, which assigns higher density to low-uncertainty states. Instantiating the classifier-guidance template by replacing $\nabla_\mathbf{x} \log p_t(y \mid \mathbf{x})$ with $\nabla_{\mathbf{x}_t} \log \tilde{p}_t(\mathbf{x}_t) = \nabla_{\mathbf{x}_t} f((\sigma_t^\theta(\mathbf{x}_t))^2)$ yields:

$$\bar{u}_{t,\mathrm{CG}}^\theta(\bar{\mathbf{x}}_t) = \bar{u}_t^\theta(\bar{\mathbf{x}}_t) + b_t w \nabla_{\bar{\mathbf{x}}_t} f((\sigma_t^\theta(\bar{\mathbf{x}}_t))^2). \quad (8)$$

The scalar function $f$ should attain larger values when the predicted velocity variance is smaller, so that the guidance term steers the trajectory toward low-uncertainty regions. In our experiments, we heuristically use the negative squared

mean across dimensions of the element-wise predicted variances, $f(\sigma^2) = -(\frac{1}{n} \sum_{i=1}^n \sigma_i^2)^2$.

**Uncertainty-aware classifier-free guidance (U-CFG).** When UA-Flow is trained with classifier-free conditioning, the CFG extrapolated mean velocity is:

$$\bar{u}_{t,\mathrm{CFG}}^\theta(\bar{\mathbf{x}}_t \mid y) = (1 + \lambda)\bar{u}_t^\theta(\bar{\mathbf{x}}_t \mid y) - \lambda \bar{u}_t^\theta(\bar{\mathbf{x}}_t \mid \varnothing), \quad (9)$$

where $\lambda \geq 0$ is the CFG scale and $\varnothing$ denotes the null condition.

Let $\sigma_{t,y}^\theta(\bar{\mathbf{x}}_t)$ and $\sigma_{t,\varnothing}^\theta(\bar{\mathbf{x}}_t)$ denote the predicted (element-wise) standard deviations of the conditional and unconditional velocities, respectively. Assuming strong correlation between the two standard deviations, we approximate the element-wise variance of the extrapolated velocity by

$$\mathrm{Var}\left[u_{t,\mathrm{CFG}}^\theta(\mathbf{x}_t \mid y)\right] \approx \left((1 + \lambda)\sigma_{t,y}^\theta(\bar{\mathbf{x}}_t) - \lambda \sigma_{t,\varnothing}^\theta(\bar{\mathbf{x}}_t)\right)^2. \quad (10)$$

We empirically show that the conditional and unconditional standard deviations are highly correlated (see Appendix G.4).

We choose $\lambda$ to minimize the total predicted variance of the extrapolated velocity with a clamp $\lambda_{\max}$ to prevent the extrapolated velocity from diverging:

$$\lambda^* = \min(\lambda_{\mathrm{opt}}, \lambda_{\max}), \quad (11)$$

where

$$\lambda_{\mathrm{opt}} = \underset{\lambda \geq 0}{\arg\min} \sum_{i=1}^n \left((1 + \lambda)\sigma_{t,y,i}^\theta(\bar{\mathbf{x}}_t) - \lambda \sigma_{t,\varnothing,i}^\theta(\bar{\mathbf{x}}_t)\right)^2. \quad (12)$$

Here, $\lambda_{\mathrm{opt}}$ admits a closed-form solution represented in Appendix C.

At each sampling step, we compute a guidance-modified mean velocity and its associated element-wise variance at the current mean state $\bar{\mathbf{x}}_t$. When enabled, U-CFG returns the guided mean and variance by extrapolating conditional and unconditional predictions and approximating the variance as in Equations (9) and (10), with the scale $\lambda^*$ computed as in Equation (11) (see Algorithm 2). When enabled, U-CG then adds the mean-velocity correction in Equation (8) (see Algorithm 3). Both mechanisms can be applied sequentially within a single sampling step.

## 4. Experiments

We design experiments to evaluate whether the uncertainty estimated by UA-Flow can be used *(i) as a sample-level reliability signal for generated samples*, and *(ii) as a control signal to improve generation via guided sampling*. To assess (i), we filter out high-uncertainty generated images and compare the resulting FID and precision/recall against baseline

*Table 1.* **Computational cost (TFLOPs) required for sampling and uncertainty quantification using the proposed method (UA-Flow) and baselines per image across datasets.** *Vanilla* only shows computational cost required for sampling.

| Method | CIFAR-10 | ImageNet-128 | ImageNet-256 |
|---|---|---|---|
| Vanilla | 7.778 | 1.097 | 4.362 |
| AU | 14.31 | 2.019 | 8.026 |
| BayesDiff | 17.72 | 2.447 | 9.731 |
| GenUnc | 46.72 | 8.493 | 33.65 |
| **UA-Flow** | **8.742** | **1.499** | **6.075** |

uncertainty estimation methods (Section 4.2). To assess (ii), we conduct controlled ablations on uncertainty-aware classifier guidance (U-CG) (Section 4.3) and uncertainty-aware classifier-free guidance (U-CFG) (Section 4.4) under matched sampling settings.

### 4.1. Experimental Setup

We evaluate our method on *CIFAR-10* (Krizhevsky et al., 2010) and ImageNet (Deng et al., 2009) at resolutions $128 \times 128$ (*ImageNet-128*) and $256 \times 256$ (*ImageNet-256*) using generative quality metrics: Fréchet Inception Distance (FID) (Heusel et al., 2017) and precision/recall (Kynkäänniemi et al., 2019). FID measures the overall distributional similarity between generated and real images, while precision and recall respectively capture sample fidelity and distributional coverage. Details not described in this subsection are provided in Appendix E.

**Training and Model Architectures.** For CIFAR-10, we use an ADM-based (Dhariwal & Nichol, 2021) unconditional flow matching model in pixel space. For ImageNet, we adopt a DiT-based conditional latent flow matching model (Dao et al., 2023) trained on images resized to $128 \times 128$ and $256 \times 256$. Specifically, we use DiT-B/2 (Peebles & Xie, 2023) as the backbone and the pretrained autoencoder from Stable Diffusion (Rombach et al., 2022) to map RGB images to a latent tensor with downsampling ratio 8 and 4 channels.

For CIFAR-10 and ImageNet-128, we first train a vanilla flow matching model and then fine-tune it by adding a variance prediction head. For ImageNet-256, we fine-tune a pretrained latent flow matching model released in prior work (Dao et al., 2023).

**Sampling Process.** All experiments use the second order Heun's method solver with 50 sampling steps. For CIFAR-10, we adopt the EDM sampling schedule (Karras et al., 2022), while for ImageNet we use uniformly spaced time steps. U-CG is applied every two sampling steps, whereas (U-)CFG is applied at every step.

### 4.2. Filtering Images with High-Uncertainty

We evaluate whether the predicted uncertainty provides a sample-level reliability signal by progressively filtering out high-uncertainty generated samples and tracking changes in FID and precision/recall.

**Baselines.** We compare against BayesDiff (Kou et al., 2023), Aleatoric Uncertainty (AU) (De Vita & Belagiannis, 2025), and Generative Uncertainty (GenUnc) (Jazbec et al., 2025). As these methods were originally proposed for diffusion models, we adapt them to flow matching by defining uncertainty over the velocity field and propagating it through the flow dynamics. Implementation details are provided in Appendix E.2.

**Uncertainty aggregation for filtering.** Uncertainty-based filtering requires assigning each generated sample a single scalar uncertainty score. GenUnc defines scalar uncertainty in the CLIP (Radford et al., 2021) embedding space, whereas UA-Flow, BayesDiff, and AU produce element-wise uncertainty maps over the image/latent tensor. For these element-wise methods, we use the final-state uncertainty map $\text{Var}[\mathbf{x}_1]$ and compute a sample-level score as the mean of the top 10% highest-uncertainty elements, which reduces the influence of large low-uncertainty background regions. UA-Flow and BayesDiff update uncertainty estimates every four sampling steps to reduce computational overhead, following the original protocol of BayesDiff. For covariance estimation in the variance propagation step shown in Equation (7), we use a single Monte Carlo sample in practice, i.e. $S = 1$. Appendix G.3.2 shows that increasing the number of probes does not noticeably improve the filtering performance.

**Filtering procedure.** Filtering experiments are conducted without classifier guidance or classifier-free guidance. We use class-conditional uncertainty for ImageNet and unconditional uncertainty for CIFAR-10. For each dataset, we generate 100k images, rank them by the sample-level uncertainty, and progressively remove the top 10% high-uncertainty samples up to 50%. At each filtering ratio, we randomly select 50k images from the remaining set and compute evaluation metrics against 50k reference images.

**Results.** Figures 2a, 6a and 6b summarize the results of uncertainty-based filtering. For UA-Flow, increasing the filtering ratio leads to higher precision and lower recall, showing a fidelity-diversity trade-off. On ImageNet, the resulting gain in sample fidelity outweighs the loss in diversity, leading to improved FID after filtering. On CIFAR-10, where the unfiltered FID is already low, this trade-off instead manifests as a slight increase in FID.

GenUnc exhibits a stronger precision-recall trade-off and achieves lower FID than UA-Flow. However, this behavior is expected, as GenUnc represents uncertainty as domain-

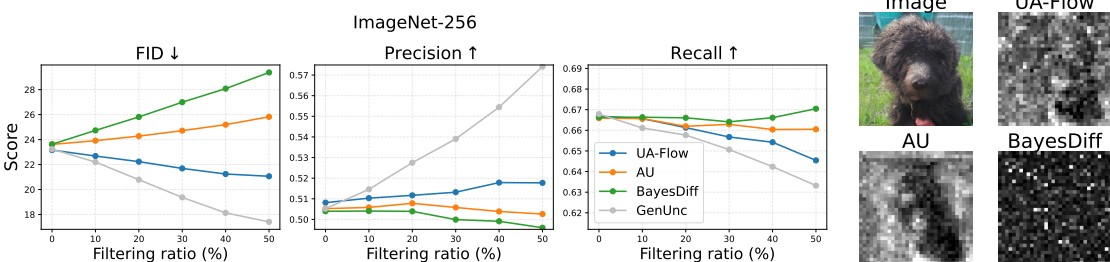

*(a)* Generative quality metrics after filtering high-uncertainty samples.

*(b)* Latent pixel-wise uncertainties.

*Figure 2.* (a) **Generative quality metrics as a function of the filtering ratio on ImageNet-256.** For each filtering level, high-uncertainty generated images are removed and 50k samples are randomly selected from the remaining set for evaluation. Compared to AU (De Vita & Belagiannis, 2025) and BayesDiff (Kou et al., 2023), which estimate element-wise uncertainty in the latent space, UA-Flow achieves lower FID and higher precision after filtering. GenUnc (Jazbec et al., 2025) is shown as a reference baseline, as it uses domain-specific scalar uncertainty estimated in the CLIP embedding space. (b) **Example latent pixel-wise uncertainty maps produced by UA-Flow, AU, and BayesDiff for the same generated image.** Brighter regions indicate higher uncertainty. For visualization, uncertainty values are normalized independently for each image.

specific scalar Gaussian entropy estimated in the CLIP embedding space, which is closely aligned with perceptual image quality. In contrast, UA-Flow estimates element-wise uncertainty directly in the latent or pixel space of the generative model, providing a finer-grained signal that correlates with the fidelity of individual image regions. As shown in Figures 2b, 6a and 6b, UA-Flow highlights spatially localized regions of high uncertainty, which is not possible for GenUnc due to its scalar formulation. Moreover, since UA-Flow does not depend on CLIP or any modality-specific embedding space, it may naturally extend to other data modalities without additional components.

In contrast, both AU and BayesDiff show increasing FID as the filtering ratio increases, indicating that their uncertainty estimates are less aligned with sample quality under flow matching. For AU, both precision and recall consistently decrease across datasets. BayesDiff exhibits a precision-recall trade-off with UA-Flow on CIFAR-10, while on ImageNet it does not show the same pattern. Qualitatively, the uncertainty maps of AU (Figures 6a and 6b) exhibit patterns that are largely inverted compared to those of UA-Flow, while BayesDiff produces noisy uncertainty maps that fail to localize high-uncertainty regions. We discuss the reason for the noisy uncertainty maps in Appendix G.1. Overall, these results suggest that the uncertainty signals produced by AU and BayesDiff do not reliably correlate with sample fidelity.

Additionally, Table 1 reports the computational overhead required to generate a single sample and quantify its uncertainty for UA-Flow and the baseline methods across datasets. UA-Flow incurs substantially lower computational overhead than all baseline uncertainty quantification methods. In particular, GenUnc requires at least $5.3\times$ higher computational cost than UA-Flow due to repeated sampling through models with different weights.

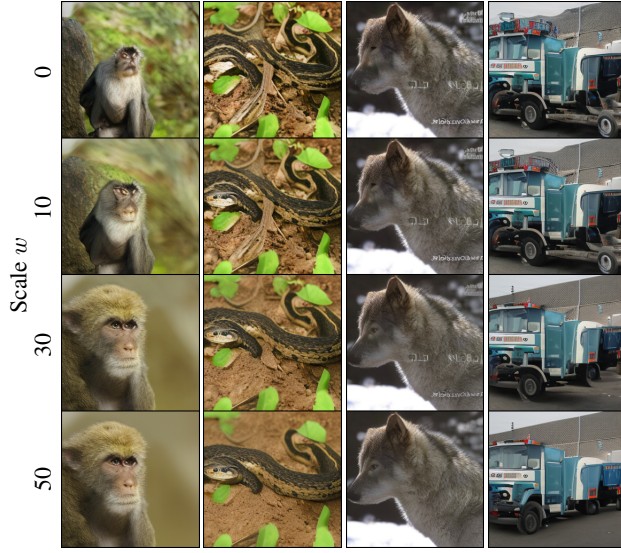

*Figure 3.* **ImageNet-256 samples under different U-CG scales $w$ at a CFG scale $\lambda = 0.5$.** Each column corresponds to a fixed class label and random seed, while rows sweep the U-CG scale $w \in \{0, 10, 30, 50\}$. As $w$ increases, samples become more class-consistent and visually simpler, reflecting the fidelity–diversity trade-off induced by steering generation toward low-uncertainty regions.

Taken together, these results demonstrate that the uncertainty estimated by UA-Flow provides a sample-level reliability signal that is not only well correlated with fidelity, but also computationally efficient.

### 4.3. Uncertainty-Aware Classifier Guidance

In this experiment, we evaluate the effect of uncertainty-aware classifier guidance (U-CG) by sweeping the U-CG

*Table 2.* **FID, precision and recall under uncertainty-aware classifier guidance (U-CG) across datasets.** For each CFG scale $\lambda$, we report the best-performing U-CG setting (lowest FID). AU denotes uncertainty-aware guidance using aleatoric uncertainty (De Vita & Belagiannis, 2025) and is included as a reference baseline. U-CG consistently improves precision compared to vanilla sampling under matched CFG settings, and it also yields better FID on ImageNet-128 and ImageNet-256.

*(a)* CIFAR-10

| Setting | U-CG | FID↓ | Precision↑ | Recall↑ |
|---|---|---|---|---|
| AU | – | 2.18 | 0.6549 | 0.6328 |
| Vanilla | – | **2.13** | 0.6570 | **0.6289** |
| U-CG only | 10 | 2.43 | **0.6585** | 0.6245 |

*(b)* ImageNet-128

| Setting | $\lambda$ | $w$ | FID↓ | Precision↑ | Recall↑ |
|---|---|---|---|---|---|
| AU | – | – | 27.21 | 0.4500 | 0.6618 |
| Vanilla | 0.0 | 0 | 27.23 | 0.4525 | **0.6697** |
| U-CG only | | 30 | **19.00** | **0.4925** | 0.6412 |
| CFG only | 0.25 | 0 | 14.76 | 0.5442 | **0.6297** |
| CFG + U-CG | | 20 | **10.71** | **0.5798** | 0.6120 |
| CFG only | 0.5 | 0 | 8.29 | 0.6251 | **0.5858** |
| CFG + U-CG | | 20 | **6.95** | **0.6452** | 0.5633 |

*(c)* ImageNet-256

| Setting | $\lambda$ | $w$ | FID↓ | Precision↑ | Recall↑ |
|---|---|---|---|---|---|
| AU | – | – | 23.14 | 0.4982 | 0.6642 |
| Vanilla | 0.0 | 0 | 23.14 | 0.4997 | **0.6619** |
| U-CG only | | 50 | **18.79** | **0.5290** | 0.6358 |
| CFG only | 0.25 | 0 | 10.31 | 0.6207 | **0.6078** |
| CFG + U-CG | | 40 | **8.65** | **0.6463** | 0.5828 |
| CFG only | 0.5 | 0 | 5.29 | 0.7154 | **0.5468** |
| CFG + U-CG | | 20 | **4.95** | **0.7286** | 0.5383 |

scale $w \in [0, 50]$ under fixed CFG scales $\lambda \in \{0, 0.25, 0.5\}$. Note that U-CG and CFG are disabled when $w = 0$ and $\lambda = 0$, respectively.

**Results.** Figure 7 shows generation quality metrics as a function of the U-CG scale. Across datasets, increasing $w$ induces a clear precision–recall trade-off: precision typically increases or peaks at an intermediate scale, while recall consistently decreases. This behavior reflects the intended effect of U-CG, which steers samples toward low-uncertainty regions of the learned velocity, favoring high-fidelity modes at the expense of diversity.

As a result, FID decreases as $w$ increases up to a dataset-dependent optimum, beyond which excessive guidance degrades performance. On CIFAR-10, where the baseline FID is already low, this trade-off may instead manifest as a slight increase in FID, consistent with the filtering behavior

*Table 3.* **FID, precision and recall under standard and uncertainty-aware classifier-free guidance (CFG and U-CFG) across ImageNet.** On ImageNet-128 and 256, we compare CFG with a scale $\lambda$ to U-CFG with a scale $\lambda^*$ capped by $\lambda_{\max}$. For each method, we report the best-performing scale (lowest FID) and the corresponding precision and recall. U-CFG achieves lower FID by retaining higher recall at the optimum.

*(a)* ImageNet-128

| | $\lambda/\lambda_{\max}$ | FID↓ | Precision↑ | Recall↑ |
|---|---|---|---|---|
| CFG | 1.0 | 5.23 | **0.7270** | 0.5031 |
| U-CFG | 1.75 | **4.82** | 0.7058 | **0.5349** |

*(b)* ImageNet-256

| | $\lambda/\lambda_{\max}$ | FID↓ | Precision↑ | Recall↑ |
|---|---|---|---|---|
| CFG | 0.75 | 4.49 | **0.7801** | 0.4933 |
| U-CFG | 1.5 | **4.35** | 0.7678 | **0.5119** |

observed in Section 4.2.

These results are summarized in Table 2, which compares generation quality with and without U-CG under matched CFG scales. For each CFG setting, we report the U-CG configuration that achieves the lowest FID. We additionally include uncertainty-aware guidance based on AU as a reference baseline. However, AU leads to only marginal changes in generation metrics under flow matching, which is consistent with the observations in Section 4.2 that its uncertainty estimates are weakly correlated with sample-level fidelity in this setting.

Figure 3 visually illustrates the qualitative effects of U-CG on ImageNet-256 samples. As the U-CG scale increases, samples become more class-consistent while exhibiting reduced background complexity, reflecting the fidelity-diversity trade-off induced by steering generation toward low-uncertainty regions. Overall, these results demonstrate that guidance based on a pseudo-likelihood derived from predicted velocity uncertainty can effectively improve sample fidelity.

### 4.4. Uncertainty-Aware Classifier-Free Guidance

We study the effect of applying uncertainty-aware classifier-free guidance (U-CFG) by evaluating FID, precision, and recall compared to CFG. For CFG, we sweep the fixed guidance scale $\lambda$. For U-CFG, since the step-wise adaptive scale $\lambda^*$ is chosen from uncertainty and clamped by a maximum $\lambda_{\max}$, we sweep $\lambda_{\max}$. We evaluate FID, precision, and recall on ImageNet-128 and 256 over a wide range of scales (e.g., $\{0, 0.25, \dots, 2.0, 3, 4, 5, 10, 15, 20\}$).

**Results.** Figures 4a and 8a compare the metrics under increasing guidance parameters $\lambda$ and $\lambda_{\max}$. As the CFG

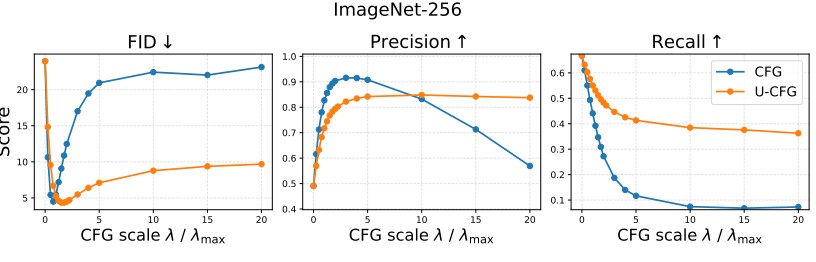
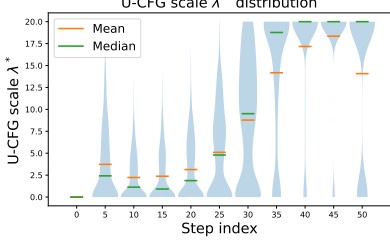

*(a)* FID, Precision, and Recall versus CFG scale $\lambda$ (CFG) or $\lambda_{\max}$ (U-CFG).

*(b)* Distribution of U-CFG scale $\lambda^*$.

*Figure 4.* (a) **FID, precision, and recall as a function of the fixed CFG scale $\lambda$ or the maximum scale $\lambda_{\max}$ of U-CFG on ImageNet-256** CFG degrades sharply at large $\lambda$, while U-CFG remains more stable as $\lambda_{\max}$ increases. (b) **Violin plots of the adaptive U-CFG scale $\lambda^*$ across sampling steps 1,000 samples.** $\lambda^*$ tends to be smaller in early steps and larger in later steps.

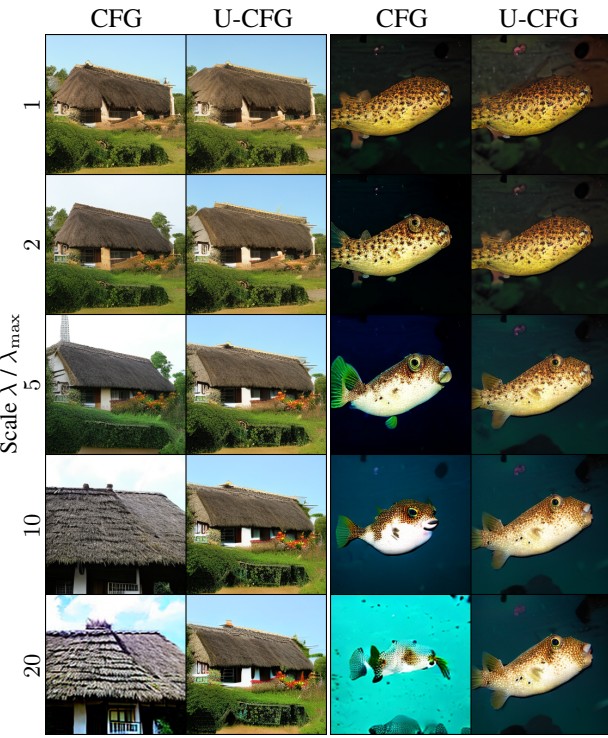

*Figure 5.* **ImageNet-256 samples under increasing CFG scale $\lambda$ and the maximum of U-CFG scale $\lambda_{\max}$.** Samples generated by standard CFG (left) and U-CFG (right) while sweeping the scale $\lambda$ (CFG) or the cap $\lambda_{\max}$ (U-CFG) in $\{1.0, 2.0, 5.0, 10, 20\}$. Large $\lambda$ in CFG can lead to oversaturation and mode collapse, whereas U-CFG better preserves global structure under large $\lambda_{\max}$ via adaptive step-wise scaling.

scale $\lambda$ grows, precision and recall degrade sharply at large $\lambda$, leading to a steep increase in FID. In contrast, precision and recall of U-CFG are changed slightly as $\lambda_{\max}$ increases, resulting in only a small FID degradation at high $\lambda_{\max}$.

This robustness can be attributed to U-CFG's adaptive, step-dependent scaling. Figures 4b and 8b show violin plots of $\lambda^*$ collected every five steps from 1,000 generated samples. We observe that $\lambda^*$ is typically smaller in early sampling

steps and becomes larger in later steps, suggesting that U-CFG avoids over-guidance when the sample is still coarse, thereby mitigating the diversity loss that is typical at high fixed CFG scales.

Qualitatively, Figure 5 shows that large $\lambda$ in standard CFG can produce oversaturated samples or mode collapse, whereas U-CFG better preserves global structure even under a large $\lambda_{\max}$ without saturation or with minimal saturation.

Finally, Table 3 reports the best-performing configurations (lowest FID) for each method. Across both datasets, U-CFG achieves lower FID than CFG despite the lower precision because it retains higher recall at the optimum. This suggests that U-CFG improves FID primarily by reducing the collapse in coverage that occurs under overly strong fixed-scale guidance. Overall, these results support that U-CFG improves generation by adaptively selecting $\lambda^*$ to reduce uncertainty at each sampling step.

## 5. Conclusion

We propose *uncertainty-aware flow matching* (UA-Flow), which predicts element-wise heteroscedastic uncertainty alongside the velocity field and propagates it through deterministic flow dynamics to obtain per-sample uncertainty, enabling uncertainty-aware classifier guidance (U-CG) and step-wise adaptive classifier-free guidance (U-CFG). Across CIFAR-10 and ImageNet, UA-Flow's uncertainty correlates more closely with sample fidelity than element-wise uncertainty quantification baselines while incurring lower computational overhead. Using this signal during sampling improves generation: U-CG induces a precision–recall trade-off by steering toward low-uncertainty modes, while U-CFG mitigates failures of large fixed guidance via adaptive scaling and remains robust under strong guidance.

Because uncertainty is predicted at the velocity level, UA-Flow is modality-agnostic and may extend beyond images, e.g., as a reliability signal in safety-critical settings such as robotics and decision-making.

## Impact Statement

This work advances uncertainty quantification for sampling-based generative models by explicitly modeling and propagating uncertainty in flow matching dynamics. By providing per-sample and spatially localized uncertainty estimates, the proposed approach can help practitioners assess the reliability of generated outputs and make more informed decisions when deploying generative models.

Potential positive impacts include improved robustness and safety in downstream applications that rely on generative models. In particular, uncertainty-aware guidance may reduce failure cases caused by overconfident or excessively guided generation.

At the same time, as with other advances in generative modeling, improved generation quality and controllability may amplify existing societal risks associated with synthetic data, including misuse, misinformation, or overreliance on automatically generated content. We emphasize that uncertainty estimates should be used as a complementary reliability signal rather than a guarantee of correctness.

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

## A. Derivation of the Uncertainty-Aware Flow Matching Loss

This appendix provides a detailed derivation of the conditional uncertainty-aware flow matching loss $\mathcal{L}_{\text{CUFM}}$ by rewriting the unconditional Gaussian negative log-likelihood loss $\mathcal{L}_{\text{UFM}}$ in terms of conditional flow matching. The key idea is to express expectations under the marginal distribution $p_t(\mathbf{x})$ using the conditional distribution $p_t(\mathbf{x} \mid \mathbf{x}_1)$, which enables tractable training despite the inaccessibility of the unconditional velocity. For notational simplicity, time is uniformly sampled, i.e. $t \sim [0, 1]$, though any alternative time-sampling distribution could be used.

We begin by expanding the uncertainty-aware flow matching loss $\mathcal{L}_{\text{UFM}}$ and decomposing it into four expectation terms, which will later be rewritten under the conditional distribution.

$$\mathcal{L}_{\text{UFM}}(\theta) = \mathbb{E}_{t,\, p_t(\mathbf{x}_t)}\Big[ \frac{\big(\bar{u}_t^\theta(\mathbf{x}_t) - u_t(\mathbf{x}_t)\big)^2}{2(\sigma_t^\theta(\mathbf{x}_t))^2} + \log(\sigma_t^\theta(\mathbf{x}_t)) \Big]$$

$$= \underbrace{\mathbb{E}_{t,\, p_t(\mathbf{x}_t)}\Big[ \frac{(\bar{u}_t^\theta(\mathbf{x}_t))^2}{2(\sigma_t^\theta(\mathbf{x}_t))^2} \Big]}_{(A)} - 2\underbrace{\mathbb{E}_{t,\, p_t(\mathbf{x}_t)}\Big[ \frac{\bar{u}_t^\theta(\mathbf{x}_t)u_t(\mathbf{x}_t)}{2(\sigma_t^\theta(\mathbf{x}_t))^2} \Big]}_{(B)} + \underbrace{\mathbb{E}_{t,\, p_t(\mathbf{x}_t)}\Big[ \frac{(u_t(\mathbf{x}_t))^2}{2(\sigma_t^\theta(\mathbf{x}_t))^2} \Big]}_{(C)} + \underbrace{\mathbb{E}_{t,\, p_t(\mathbf{x}_t)}\Big[ \log(\sigma_t^\theta(\mathbf{x}_t)) \Big]}_{(D)} \tag{13}$$

Term (A) depends only on the predicted mean and variance and can be rewritten by expressing the marginal distribution $p_t(\mathbf{x}_t)$ as an integral over the conditional distribution $p_t(\mathbf{x}_t \mid \mathbf{x}_1)$ and the data distribution $p_1(\mathbf{x}_1)$.

$$(A) = \int \frac{(\bar{u}_t^\theta(\mathbf{x}_t))^2}{2(\sigma_t^\theta(\mathbf{x}_t))^2} p_t(\mathbf{x}_t) d\mathbf{x}_t dt = \int \frac{(\bar{u}_t^\theta(\mathbf{x}_t))^2}{2(\sigma_t^\theta(\mathbf{x}_t))^2} p_t(\mathbf{x}_t \mid \mathbf{x}_1)p_1(\mathbf{x}_1) d\mathbf{x}_1 d\mathbf{x}_t dt = \mathbb{E}_{t, p_1(\mathbf{x}_1), p_t(\mathbf{x}_t \mid \mathbf{x}_1)}\Big[ \frac{(\bar{u}_t^\theta(\mathbf{x}_t))^2}{2(\sigma_t^\theta(\mathbf{x}_t))^2} \Big] \tag{14}$$

Term (B) involves the cross term between the predicted mean and the true unconditional velocity. Since the unconditional velocity $u_t(\mathbf{x})$ is intractable, we rewrite it using the law of total expectation under the conditional flow matching formulation.

$$\begin{aligned}
(B) &= \int \frac{\bar{u}_t^\theta(\mathbf{x}_t)u_t(\mathbf{x}_t)}{2(\sigma_t^\theta(\mathbf{x}_t))^2} p_t(\mathbf{x}_t) d\mathbf{x}_t dt \\
&= \int \frac{\bar{u}_t^\theta(\mathbf{x}_t)}{2(\sigma_t^\theta(\mathbf{x}_t))^2}\Big( \int \frac{u_t(\mathbf{x}_t \mid \mathbf{x}_1)p_t(\mathbf{x}_t \mid \mathbf{x}_1)p_1(\mathbf{x}_1)}{p_t(\mathbf{x}_t)} d\mathbf{x}_1 \Big) p_t(\mathbf{x}_t) d\mathbf{x}_t dt \\
&= \int \frac{\bar{u}_t^\theta(\mathbf{x}_t)u_t(\mathbf{x}_t \mid \mathbf{x}_1)}{2(\sigma_t^\theta(\mathbf{x}_t))^2} p_t(\mathbf{x}_t \mid \mathbf{x}_1)p_1(\mathbf{x}_1) d\mathbf{x}_1 d\mathbf{x}_t dt \\
&= \mathbb{E}_{t, p_1(\mathbf{x}_1), p_t(\mathbf{x}_t \mid \mathbf{x}_1)}\Big[ \frac{\bar{u}_t^\theta(\mathbf{x}_t)u_t(\mathbf{x}_t \mid \mathbf{x}_1)}{2(\sigma_t^\theta(\mathbf{x}_t))^2} \Big]
\end{aligned} \tag{15}$$

By the same change of measure, terms (C) and (D) can be rewritten as

$$(C) = \mathbb{E}_{t, p_1(\mathbf{x}_1), p_t(\mathbf{x}_t \mid \mathbf{x}_1)}\Big[ \frac{(u_t(\mathbf{x}_t))^2}{2(\sigma_t^\theta(\mathbf{x}_t))^2} \Big] \tag{16}$$

$$(D) = \mathbb{E}_{t, p_1(\mathbf{x}_1), p_t(\mathbf{x}_t \mid \mathbf{x}_1)}\Big[ \log(\sigma_t^\theta(\mathbf{x}_t)) \Big] \tag{17}$$

Therefore, $\mathcal{L}_{UFM}(\theta)$ can be rewritten as:

$$\begin{aligned}
\mathcal{L}_{\text{UFM}}(\theta) &= \mathbb{E}_{t, p_1(\mathbf{x}_1), p_t(\mathbf{x}_t \mid \mathbf{x}_1)}\Big[ \frac{(\bar{u}_t^\theta(\mathbf{x}_t))^2 - 2\bar{u}_t^\theta(\mathbf{x}_t)u_t(\mathbf{x}_t \mid \mathbf{x}_1) + (u_t(\mathbf{x}_t))^2}{2(\sigma_t^\theta(\mathbf{x}_t))^2} + \log(\sigma_t^\theta(\mathbf{x}_t)) \Big] \\
&= \mathbb{E}_{t, p_1(\mathbf{x}_1), p_t(\mathbf{x}_t \mid \mathbf{x}_1)}\Big[ \frac{(\bar{u}_t^\theta(\mathbf{x}_t) - u_t(\mathbf{x}_t \mid \mathbf{x}_1))^2 + (u_t(\mathbf{x}_t))^2 - (u_t(\mathbf{x}_t \mid \mathbf{x}_1))^2}{2(\sigma_t^\theta(\mathbf{x}_t))^2} + \log(\sigma_t^\theta(\mathbf{x}_t)) \Big]
\end{aligned} \tag{18}$$

However, we cannot evaluate the true unconditional flow $u_t(\mathbf{x}_t)$ in closed form. Using the identity in Equation (19), we can rewrite $u_t(\mathbf{x}_t)$ as a ratio of expectations over $\mathbf{x}_1 \sim p_1$:

$$u_t(\mathbf{x}_t) = \int \frac{u_t(\mathbf{x}_t \mid \mathbf{x}_1)p_t(\mathbf{x}_t \mid \mathbf{x}_1)p_1(\mathbf{x}_1)}{p_t(\mathbf{x}_t)} d\mathbf{x}_1 = \frac{\int u_t(\mathbf{x}_t \mid \mathbf{x}_1)p_t(\mathbf{x}_t \mid \mathbf{x}_1)p_1(\mathbf{x}_1)d\mathbf{x}_1}{\int p_t(\mathbf{x}_t \mid \mathbf{x}_1)p_1(\mathbf{x}_1)d\mathbf{x}_1} = \frac{\mathbb{E}_{p_1(\mathbf{x}_1)}[u_t(\mathbf{x}_t \mid \mathbf{x}_1)p_t(\mathbf{x}_t \mid \mathbf{x}_1)]}{\mathbb{E}_{p_1(\mathbf{x}_1)}[p_t(\mathbf{x}_t \mid \mathbf{x}_1)]}. \tag{19}$$

This suggests a self-normalized importance-sampling estimator based on a mini-batch $\{\mathbf{x}_{1,b}\}_{b=1}^{B} \sim p_1$:

$$\hat{u}_t(\mathbf{x}_t) = \frac{\sum_{b=1}^{B} u_t(\mathbf{x}_t \mid \mathbf{x}_{1,b}) \, p_t(\mathbf{x}_t \mid \mathbf{x}_{1,b})}{\sum_{b=1}^{B} p_t(\mathbf{x}_t \mid \mathbf{x}_{1,b})}. \tag{20}$$

Since $u_t(\mathbf{x}_t)$ is intractable, we approximate $(u_t(\mathbf{x}_t))^2$ by $(\hat{u}_t(\mathbf{x}_t))^2$ in our objective. This ratio-of-expectations naturally motivates an importance-weighted approximation, providing a tractable proxy for the unconditional target. Substituting $\hat{u}_t(\mathbf{x}_t)^2$ yields the correction term $U_t(\mathbf{x}_t, \mathbf{x}_1) := \hat{u}_t(\mathbf{x}_t)^2 - u_t(\mathbf{x}_t \mid \mathbf{x}_1)^2$ and results in the conditional objective $\mathcal{L}_{\text{CUFM}}(\theta)$ in Equation (2).

**Remark on Jensen bias in approximating $(u_t(\mathbf{x}_t))^2$ and why we still keep $U_t(\mathbf{x}_t, \mathbf{x}_1)$.** However, $\hat{u}_t(\mathbf{x}_t)^2$ is not an unbiased estimator of $u_t(\mathbf{x}_t)^2$ even when $\hat{u}_t(\mathbf{x}_t)$ is a consistent proxy for $u_t(\mathbf{x}_t)$. This follows from the identity $\mathbb{E}[\hat{u}^2] - (\mathbb{E}[\hat{u}])^2 = \text{Var}(\hat{u})$, i.e., squaring introduces a Jensen gap proportional to the estimator variance. Accordingly, the bias is controlled by the (mini-batch) estimator variance and typically decreases as the mini-batch size $B$ increases.

Despite this limitation, introducing the correction term $U_t(\mathbf{x}_t, \mathbf{x}_1) := \hat{u}_t(\mathbf{x}_t)^2 - u_t(\mathbf{x}_t \mid \mathbf{x}_1)^2$ still yields a closer surrogate to the original unconditional objective than omitting $U_t$ altogether. Indeed, letting $p_{t|1}(\mathbf{x}_1 \mid \mathbf{x}_t) = \frac{p_t(\mathbf{x}_t \mid \mathbf{x}_1) \, p_1(\mathbf{x}_1)}{p_t(\mathbf{x}_t)}$ denote the induced posterior in Equation (19), we have $u_t(\mathbf{x}_t) = \mathbb{E}_{p_{t|1}(\mathbf{x}_1 \mid \mathbf{x}_t)}[u_t(\mathbf{x}_t \mid \mathbf{x}_1)]$ and thus

$$\mathbb{E}_{p_{t|1}(\mathbf{x}_1 \mid \mathbf{x}_t)}\left[u_t(\mathbf{x}_t)^2 - u_t(\mathbf{x}_t \mid \mathbf{x}_1)^2\right] = -\text{Var}_{p_{t|1}(\mathbf{x}_1 \mid \mathbf{x}_t)}(u_t(\mathbf{x}_t \mid \mathbf{x}_1)) \leq 0.$$

Therefore, dropping $U_t$ implicitly sets this negative term to zero, incurring a systematic bias that does not vanish with $B$. In contrast, our proxy retains this variance-related correction up to the residual bias in $\hat{u}_t(\mathbf{x}_t)^2$, which diminishes as the mini-batch estimator variance decreases (e.g., as $B$ increases).

## B. Details on Variance Propagation and Covariance Approximations

This appendix provides detailed derivations for the variance evolution equations and tractable approximations of the covariance between the state and the velocity field.

### B.1. Derivation of Equation (4)

Using the Gaussian velocity model $u_t^\theta(\mathbf{x}) = \bar{u}_t^\theta(\mathbf{x}_t) + \sigma_t^\theta(\mathbf{x}_t) \odot \epsilon$ with $\epsilon \sim \mathcal{N}(0, I)$, we have $\mathbb{E}[u_t^\theta(\mathbf{x}_t)] = \mathbb{E}[\bar{u}_t^\theta(\mathbf{x}_t)]$. Applying a first-order Taylor expansion of $\bar{u}_t^\theta(\mathbf{x}_t)$ around $\bar{\mathbf{x}}_t := \mathbb{E}[\mathbf{x}_t]$ yields

$$\frac{d\bar{\mathbf{x}}_t}{dt} = \mathbb{E}\left[u_t^\theta(\mathbf{x}_t)\right] = \mathbb{E}\left[\bar{u}_t^\theta(\mathbf{x}_t)\right] \approx \mathbb{E}\left[\bar{u}_t^\theta(\bar{\mathbf{x}}_t) + J_t^\theta(\bar{\mathbf{x}}_t)(\mathbf{x}_t - \bar{\mathbf{x}}_t)\right] = \bar{u}_t^\theta(\bar{\mathbf{x}}_t). \tag{21}$$

### B.2. Derivation of Equation (5)

Under Euler integration between times $t$ and $t + \Delta t$, the flow dynamics become

$$\mathbf{x}_{t+\Delta t} = \mathbf{x}_t + u_t^\theta(\mathbf{x}_t) \, \Delta t. \tag{22}$$

Applying the element-wise variance identity $\text{Var}(X + aY) = \text{Var}(X) + a^2\text{Var}(Y) + 2a\,\text{Cov}(X, Y)$ to Equation (22) yields Equation (5), where $Y = u_t^\theta(\mathbf{x}_t)$.

Next, we justify the approximation $\text{Var}(u_t^\theta(\mathbf{x}_t)) \approx (\sigma_t^\theta(\bar{\mathbf{x}}_t))^2$ used in the main text. By the law of total variance and $u_t^\theta(\mathbf{x}_t) = \bar{u}_t^\theta(\mathbf{x}_t) + \sigma_t^\theta(\mathbf{x}_t) \odot \epsilon$ with $\epsilon \sim \mathcal{N}(0, I)$, we have

$$\begin{aligned}
\text{Var}(u_t^\theta(\mathbf{x}_t)) &= \mathbb{E}[\text{Var}(u_t^\theta(\mathbf{x}_t) \mid \mathbf{x}_t)] + \text{Var}(\mathbb{E}[u_t^\theta(\mathbf{x}_t) \mid \mathbf{x}_t]) \\
&= \mathbb{E}_{\mathbf{x}_t}\left[(\sigma_t^\theta(\mathbf{x}_t))^2\right] + \text{Var}(\bar{u}_t^\theta(\mathbf{x}_t)) \\
&\approx \mathbb{E}_{\mathbf{x}_t}\left[(\sigma_t^\theta(\bar{\mathbf{x}}_t))^2 + \frac{\partial(\sigma_t^\theta)^2}{\partial\mathbf{x}}\Big|_{\bar{\mathbf{x}}_t}(\mathbf{x}_t - \bar{\mathbf{x}}_t)\right] + \text{Var}(\bar{u}_t^\theta(\mathbf{x}_t)) \\
&= (\sigma_t^\theta(\bar{\mathbf{x}}_t))^2 + \text{Var}(\bar{u}_t^\theta(\mathbf{x}_t))
\end{aligned} \tag{23}$$

where the last step follows by dropping higher-order terms and using $\mathbb{E}[\mathbf{x}_t - \bar{\mathbf{x}}_t] = 0$. The remaining term $\mathrm{Var}(\bar{u}_t^\theta(\mathbf{x}_t))$ captures variance induced by the spread of $\mathbf{x}_t$. In principle, it can be estimated by Monte Carlo sampling: draw $\mathbf{x}_{t,i} \sim \mathcal{N}(\bar{\mathbf{x}}_t, \mathrm{Var}[\mathbf{x}_t])$ and compute the empirical variance of $\bar{u}_t^\theta(\mathbf{x}_{t,i})$.

In the main text, we focus on the predicted heteroscedastic uncertainty and avoid this additional Monte Carlo overhead. Thus, we neglect $\mathrm{Var}(\bar{u}_t^\theta(\mathbf{x}_t))$ and use the approximation

$$\mathrm{Var}(u_t^\theta(\mathbf{x}_t)) \approx (\sigma_t^\theta(\bar{\mathbf{x}}_t))^2.$$

Appendix G.3.1 further shows that explicitly including $\mathrm{Var}(\bar{u}_t^\theta(\mathbf{x}_t))$ has negligible empirical effect on the resulting uncertainty estimates.

## B.3. Derivation of Equation (6)

We write $u_t^\theta(\mathbf{x}_t) = \bar{u}_t^\theta(\mathbf{x}_t) + \sigma_t^\theta(\mathbf{x}_t) \odot \epsilon$ with $\epsilon \sim \mathcal{N}(0, I)$. The noise term does not contribute to the covariance because $\epsilon$ is independent of $\mathbf{x}_t$ and has zero mean. Thus,

$$\begin{aligned}
\mathrm{Cov}(\mathbf{x}_t, u_t^\theta(\mathbf{x}_t)) &= \mathbb{E}[\mathbf{x}_t \odot u_t^\theta(\mathbf{x}_t)] - \mathbb{E}[\mathbf{x}_t] \odot \mathbb{E}[u_t^\theta(\mathbf{x}_t)] \\
&= \mathbb{E}_{\mathbf{x}_t}[\mathbf{x}_t \odot \bar{u}_t^\theta(\mathbf{x}_t)] - \bar{\mathbf{x}}_t \odot \mathbb{E}_{\mathbf{x}_t}[\bar{u}_t^\theta(\mathbf{x}_t)] \\
&\approx \mathbb{E}_{\mathbf{x}_t}[\mathbf{x}_t \odot \bar{u}_t^\theta(\mathbf{x}_t)] - \bar{\mathbf{x}}_t \odot \bar{u}_t^\theta(\bar{\mathbf{x}}_t).
\end{aligned} \quad (24)$$

Applying a first-order Taylor expansion of $\bar{u}_t^\theta(\mathbf{x}_t)$ around $\bar{\mathbf{x}}_t$ gives

$$\begin{aligned}
\mathbb{E}_{\mathbf{x}_t}\left[\mathbf{x}_t \odot \bar{u}_t^\theta(\mathbf{x}_t)\right] &\approx \mathbb{E}_{\mathbf{x}_t}\left[\mathbf{x}_t \odot \left(\bar{u}_t^\theta(\bar{\mathbf{x}}_t) + J_t^\theta(\bar{\mathbf{x}}_t)(\mathbf{x}_t - \bar{\mathbf{x}}_t)\right)\right] \\
&= \bar{\mathbf{x}}_t \odot \bar{u}_t^\theta(\bar{\mathbf{x}}_t) + \mathbb{E}_{\mathbf{x}_t}\left[(\mathbf{x}_t - \bar{\mathbf{x}}_t) \odot \left(J_t^\theta(\bar{\mathbf{x}}_t)(\mathbf{x}_t - \bar{\mathbf{x}}_t)\right)\right].
\end{aligned} \quad (25)$$

To keep the propagation tractable in high dimensions, we approximate $\mathrm{Cov}(\mathbf{x}_t) \in \mathbb{R}^{n \times n}$ as diagonal (i.e., we neglect off-diagonal entries). Then the $i$-th element of the last expectation becomes

$$\begin{aligned}
\mathbb{E}\left[(\mathbf{x}_t - \bar{\mathbf{x}}_t)_i \left(J_t^\theta(\bar{\mathbf{x}}_t)(\mathbf{x}_t - \bar{\mathbf{x}}_t)\right)_i\right] &= \sum_{j=1}^{n} (J_t^\theta(\bar{\mathbf{x}}_t))_{ij}\, \mathbb{E}\left[(\mathbf{x}_t - \bar{\mathbf{x}}_t)_i\,(\mathbf{x}_t - \bar{\mathbf{x}}_t)_j\right] \\
&= \sum_{j=1}^{n} (J_t^\theta(\bar{\mathbf{x}}_t))_{ij}\,(\mathrm{Cov}(\mathbf{x}_t))_{ij} = (J_t^\theta(\bar{\mathbf{x}}_t))_{ii}\,\mathrm{Var}[\mathbf{x}_t]_i.
\end{aligned} \quad (26)$$

Therefore,

$$\mathbb{E}_{\mathbf{x}_t}\left[\mathbf{x}_t \odot \bar{u}_t^\theta(\mathbf{x}_t)\right] \approx \bar{\mathbf{x}}_t \odot \bar{u}_t^\theta(\bar{\mathbf{x}}_t) + \mathrm{diag}(J_t^\theta(\bar{\mathbf{x}}_t)) \odot \mathrm{Var}[\mathbf{x}_t]. \quad (27)$$

Combining Equations (24) to (27) yields Equation (6).

## B.4. Derivation of Equation (7)

Let $\mathbf{r} \in \mathbb{R}^n$ be a Rademacher vector with independent entries sampled uniformly from $\{-1, +1\}$, so that $\mathbb{E}[r_i r_j] = \delta_{ij}$. Define $\boldsymbol{\sigma}_t^x := \sqrt{\mathrm{Var}[\mathbf{x}_t]}$ and $\mathbf{v} := \boldsymbol{\sigma}_t^x \odot \mathbf{r}$. Then, for each coordinate $i$,

$$\mathbb{E}\left[v_i (J_t^\theta(\bar{\mathbf{x}}_t)\mathbf{v})_i\right] = \sum_{j=1}^{n} (J_t^\theta(\bar{\mathbf{x}}_t))_{ij}\, \mathbb{E}[v_i v_j] = (J_t^\theta(\bar{\mathbf{x}}_t))_{ii}\,(\boldsymbol{\sigma}_t^x)_i^2,$$

since $\mathbb{E}[v_i v_j] = (\boldsymbol{\sigma}_t^x)_i (\boldsymbol{\sigma}_t^x)_j\, \mathbb{E}[r_i r_j]$ and $\mathbb{E}[r_i r_j] = \delta_{ij}$. Stacking all coordinates gives

$$\mathbb{E}\left[\mathbf{v} \odot (J_t^\theta(\bar{\mathbf{x}}_t)\mathbf{v})\right] = \mathrm{diag}(J_t^\theta(\bar{\mathbf{x}}_t)) \odot \mathrm{Var}[\mathbf{x}_t]. \quad (28)$$

Therefore, Equation (7) provides an unbiased Monte Carlo estimator of $\mathrm{diag}(J_t^\theta(\bar{\mathbf{x}}_t)) \odot \mathrm{Var}[\mathbf{x}_t]$ using only Jacobian–vector products.

**B.5. Approximations of** $\mathrm{Cov}(\mathbf{x}_t, u_t^\theta(\mathbf{x}_t))$

We approximate $\mathrm{Cov}(\mathbf{x}_t, u_t^\theta(\mathbf{x}_t))$ in three tractable ways:

$$
\mathrm{Cov}(\mathbf{x}_t, u_t^\theta(\mathbf{x}_t)) \approx
\begin{cases}
0 & \text{(Option 1)} \\[2mm]
\frac{1}{S} \sum_{i=1}^S (\boldsymbol{\sigma}_t^x \odot \mathbf{r}_i) \odot \left( J_t^\theta(\bar{\mathbf{x}}_t)(\boldsymbol{\sigma}_t^x \odot \mathbf{r}_i) \right) & \text{(Option 2)} \\[2mm]
\frac{1}{S} \sum_{i=1}^S \mathbf{x}_{t,i} \odot \bar{u}_t^\theta(\mathbf{x}_{t,i}) - \bar{\mathbf{x}}_t \odot \left( \frac{1}{S} \sum_{i=1}^S \bar{u}_t^\theta(\mathbf{x}_{t,i}) \right) & \text{(Option 3)}
\end{cases}
\tag{29}
$$

where $\mathbf{r}_i$ are Rademacher vectors and $\mathbf{x}_{t,i} \sim \mathcal{N}(\bar{\mathbf{x}}_t, \mathrm{diag}(\mathrm{Var}[\mathbf{x}_t]))$. Here $S$ denotes the number of samples.

**Option 1.** Ignore the covariance term. This is the cheapest choice computationally, but it discards the interaction between state and flow.

**Option 2.** This is the estimator deployed in our implementation. It estimates $\mathrm{diag}(J_t^\theta(\bar{\mathbf{x}}_t)) \odot \mathrm{Var}[\mathbf{x}_t]$ using $S$ Rademacher probes and Jacobian-vector products; see Algorithm 1 for details.

**Option 3.** A Monte Carlo alternative, similar to BayesDiff (Kou et al., 2023), draws $\mathbf{x}_{t,i} \sim \mathcal{N}(\bar{\mathbf{x}}_t, \mathrm{Var}[\mathbf{x}_t])$ for $i = 1, \cdots, S$ and estimates the covariance directly from sample moments.

# C. Derivations of $\lambda^*$

Recall that U-CFG chooses a scalar CFG scale $\lambda \geq 0$ by minimizing the total predicted variance of the extrapolated velocity. Using the notation in Section 3.3, let $\sigma_{t,y}^\theta(\bar{\mathbf{x}}_t) \in \mathbb{R}^n$ and $\sigma_{t,\varnothing}^\theta(\bar{\mathbf{x}}_t) \in \mathbb{R}^n$ denote the element-wise standard deviations of the conditional and unconditional velocities, respectively. As described in Equation (10), the element-wise standard deviation of the extrapolated velocity is

$$
\tilde{\sigma}(\lambda) = (1+\lambda)\sigma_{t,y}^\theta(\bar{\mathbf{x}}_t) - \lambda \sigma_{t,\varnothing}^\theta(\bar{\mathbf{x}}_t) = \sigma_{t,y}^\theta(\bar{\mathbf{x}}_t) + \lambda(\sigma_{t,y}^\theta(\bar{\mathbf{x}}_t) - \sigma_{t,\varnothing}^\theta(\bar{\mathbf{x}}_t)).
\tag{30}
$$

Therefore, Equation (12) can be written as the following least-squares problem:

$$
\lambda_{\mathrm{opt}} = \arg\min_{\lambda \geq 0} \sum_{i=1}^n \tilde{\sigma}_i(\lambda)^2 = \arg\min_{\lambda \geq 0} \|\sigma_{t,y}^\theta(\bar{\mathbf{x}}_t) + \lambda \mathbf{d}\|_2^2, \qquad \mathbf{d} := \sigma_{t,y}^\theta(\bar{\mathbf{x}}_t) - \sigma_{t,\varnothing}^\theta(\bar{\mathbf{x}}_t).
\tag{31}
$$

Expanding the objective yields a convex quadratic in $\lambda$:

$$
J(\lambda) = \|\sigma_{t,y}^\theta(\bar{\mathbf{x}}_t) + \lambda \mathbf{d}\|_2^2 = \sigma_{t,y}^\theta(\bar{\mathbf{x}}_t)^\top \sigma_{t,y}^\theta(\bar{\mathbf{x}}_t) + 2\lambda \, \mathbf{d}^\top \sigma_{t,y}^\theta(\bar{\mathbf{x}}_t) + \lambda^2 \mathbf{d}^\top \mathbf{d}.
\tag{32}
$$

When $\mathbf{d}^\top \mathbf{d} > 0$, the unconstrained minimizer is obtained by setting $\frac{dJ}{d\lambda} = 0$:

$$
\frac{dJ}{d\lambda} = 2\, \mathbf{d}^\top \sigma_{t,y}^\theta(\bar{\mathbf{x}}_t) + 2\lambda \, \mathbf{d}^\top \mathbf{d} = 0 \quad \implies \quad \lambda^\star = -\frac{\mathbf{d}^\top \sigma_{t,y}^\theta(\bar{\mathbf{x}}_t)}{\mathbf{d}^\top \mathbf{d}} = \frac{(\sigma_{t,\varnothing}^\theta)^\top \sigma_{t,y}^\theta - \|\sigma_{t,y}^\theta\|_2^2}{\|\sigma_{t,y}^\theta - \sigma_{t,\varnothing}^\theta\|_2^2}.
\tag{33}
$$

Imposing the constraint $\lambda \geq 0$ gives the closed-form solution:

$$
\lambda_{\mathrm{opt}} = \max(0, \lambda^\star).
\tag{34}
$$

If $\mathbf{d}^\top \mathbf{d} = 0$ (i.e., $\boldsymbol{\sigma}_y = \boldsymbol{\sigma}_\varnothing$), the objective $J(\lambda)$ is constant in $\lambda$, and we set $\lambda_{\mathrm{opt}} = 0$. Finally, we apply the clamp in Equation (11):

$$
\lambda^* = \min(\lambda_{\mathrm{opt}}, \lambda_{\max}).
$$

# D. Algorithms

This section summarizes the key sampling-time procedures used by UA-Flow. Algorithm 1 provides a Hutchinson's diagonal estimator (Bekas et al., 2007; Dharangutte & Musco, 2023) of $\mathrm{diag}(J_t^\theta(\bar{\mathbf{x}}_t)) \odot \mathrm{Var}[\mathbf{x}_t]$ via Jacobian-vector products(JVP)-based covariance approximation in Equation (7). Algorithm 2 gives uncertainty-aware classifier-free guidance (U-CFG), which computes the guided mean and variance $(\bar{u}, \sigma^2)$ by combining conditional and unconditional predictions using an adaptive guidance scale $\lambda^*$. Algorithm 3 gives uncertainty-aware classifier guidance (U-CG), which corrects the mean velocity using the gradient of a guidance objective defined on the predicted uncertainty.

---

**Algorithm 1** Estimating $\mathrm{diag}(J_t^\theta(\bar{\mathbf{x}}_t)) \odot \mathrm{Var}[\mathbf{x}_t]$

---

**Input:** velocity mean $\bar{u}_t^\theta(\cdot)$, state mean $\bar{\mathbf{x}}_t$, state variance $\mathrm{Var}[\mathbf{x}_t]$, probes $S$
$\boldsymbol{\sigma}_t^x \leftarrow \sqrt{\mathrm{Var}[\mathbf{x}_t]}, \quad \mathbf{g} \leftarrow \mathbf{0}$
**for** $k = 1, \ldots, S$ **do**
    Sample $\mathbf{r}^{(k)} \in \{\pm 1\}^n$
    $\mathbf{v}^{(k)} \leftarrow \boldsymbol{\sigma}_t^x \odot \mathbf{r}^{(k)}$
    $\mathbf{J}\mathbf{v}^{(k)} \leftarrow \mathrm{JVP}(\bar{u}_t^\theta, \bar{\mathbf{x}}_t, \mathbf{v}^{(k)})$
    $\mathbf{g} \leftarrow \mathbf{g} + \mathbf{v}^{(k)} \odot \mathbf{J}\mathbf{v}^{(k)}$
**end for**
**return** $\mathbf{g}/S$

---

**Algorithm 2** Uncertainty-aware classifier-free guidance (U-CFG): Guided mean and variance $(\bar{u}, \sigma^2)$

---

**Input:** velocity mean and variance $(\bar{u}_t^\theta, (\sigma_t^\theta)^2)$, state mean $\bar{\mathbf{x}}_t$, condition $y$, maximum CFG scale $\lambda_{\max}$
$(\bar{u}_y, \sigma_y^2) \leftarrow (\bar{u}_t^\theta(\bar{\mathbf{x}}_t \mid y), (\sigma_t^\theta(\bar{\mathbf{x}}_t \mid y))^2)$
$(\bar{u}_\varnothing, \sigma_\varnothing^2) \leftarrow (\bar{u}_t^\theta(\bar{\mathbf{x}}_t \mid \varnothing), (\sigma_t^\theta(\bar{\mathbf{x}}_t \mid \varnothing))^2)$
Compute $\lambda_{\mathrm{opt}}$                                              ▷ Appendix C
$\lambda^* \leftarrow \min(\lambda_{\mathrm{opt}}, \lambda_{\max})$
$\bar{u} \leftarrow (1 + \lambda^*)\bar{u}_y - \lambda^*\bar{u}_\varnothing$
$\sigma^2 \leftarrow ((1 + \lambda^*)\sigma_y - \lambda^*\sigma_\varnothing)^2$
**return** $(\bar{u}, \sigma^2)$

---

## E. Implementation Details

### E.1. Model & Training

*Table 4.* **Hyper-parameters.** When two learning rates are reported as $(\eta_{\mathrm{head}}, \eta_{\mathrm{backbone}})$, they denote the head and backbone learning rates, respectively.

| | CIFAR-10 | | ImageNet-128 | | ImageNet-256 |
| | pretraining | finetuning | pretraining | finetuning | finetuning |
|---|---|---|---|---|---|
| Learning rate | $1e\text{-}4$ | $(1e\text{-}5, 1e\text{-}6)$ | $2e\text{-}4$ | $(2e\text{-}5, 2e\text{-}5)$ | $(1e\text{-}5, 1e\text{-}5)$ |
| AdamW $(\beta_1, \beta_2)$ | $(0.9, 0.95)$ | $(0.9, 0.999)$ | $(0.9, 0.999)$ | $(0.9, 0.999)$ | $(0.9, 0.999)$ |
| Gradient warm-up step | – | 10000 | – | 1000 | 1000 |
| Gradient clipping | – | 1.0 | – | 1.0 | 1.0 |
| EMA decay rate | 0.999 | 0.999 | 0.9999 | 0.9999 | 0.9999 |
| Batch size | 128 | 128 | 1024 | 1024 | 512 |
| Epochs | 1000 | 100 | 900 | 100 | 90 |
| GPUs | 2 | 2 | 4 | 4 | 4 |

**Architectures.** For CIFAR-10, we use an ADM-based unconditional flow matching model. The architectural configuration follows standard ADM design choices and is detailed in Table 5. For ImageNet-128 and ImageNet-256, we adopt DiT-based conditional latent flow matching models, as described in Section 4, using DiT-B/2 as the backbone and a pretrained autoencoder to map RGB images to a latent space.

**Training procedure.** We summarize the training hyper-parameters for all datasets in Table 4. All models are trained with EMA, using the decay rates reported in the table. For CIFAR-10, we employ a skewed time-step sampling strategy used in EDM (Karras et al., 2022) during training. For ImageNet-256, we train the model using reverse-time parameterization, consistent with the pretrained latent flow matching setup.

For datasets involving fine-tuning, we apply linear gradient warm-up for the specified number of steps and use gradient clipping with a maximum norm of 1.0. Batch sizes and the number of GPUs used for each setting are reported in Table 4.

---

**Algorithm 3** Uncertainty-aware classifier guidance (U-CG): Mean velocity correction

---

**Input:** state mean $\bar{\mathbf{x}}_t$, current $(\bar{u}, \sigma^2)$, scales $w$, schedule $b_t$, guidance function $f$
$\mathbf{d} \leftarrow \nabla_{\bar{\mathbf{x}}_t} f(\sigma^2)$
$\bar{u} \leftarrow \bar{u} + b_t w \mathbf{d}$
**return** $\bar{u}$

---

*Table 5.* ADM configuration for CIFAR-10.

|  | CIFAR-10 |
|---|---|
| # of ResNet blocks per scale | 4 |
| Base channels | 128 |
| Channel multiplier per scale | (2, 2, 2) |
| Attention resolutions | 2 |
| Dropout | 0.3 |

For the reweighted mini-batch estimator used to compute $\hat{u}_t$, the batch size is defined per GPU and aggregated across devices during training.

**Uncertainty modeling.** For uncertainty-aware training, we add a variance prediction head to the pretrained flow matching backbone. The variance head predicts $\log \sigma$ to improve numerical stability during training. We use the $\beta$-NLL (Seitzer et al., 2022) objective with $\beta = 1.0$ for all experiments.

For ImageNet-128 and ImageNet-256, classifier-free conditioning is enabled by applying label dropout with probability 0.1 during training.

### E.2. Flow Matching Versions of Baselines

Existing uncertainty quantification methods for sampling-based generative models are primarily developed for diffusion models. To enable a fair comparison with UA-Flow, we adapt these baselines (De Vita & Belagiannis, 2025; Kou et al., 2023; Jazbec et al., 2025) to the flow matching framework. In this subsection, we describe how the diffusion-based formulations of these methods are converted into their flow matching counterparts.

**Aleatoric Uncertainty (AU) (De Vita & Belagiannis, 2025).** In the original formulation, Aleatoric Uncertainty (AU) estimates sample uncertainty using the score function of a diffusion model. At each sampling step $t$, the model first estimates the clean data sample $\hat{\mathbf{x}}_1$ from the predicted score $\epsilon_t^\theta(\mathbf{x}_t)$. Multiple perturbed states $\hat{\mathbf{x}}_t^i$ are then generated by re-noising the estimated data, and the variance of the score predictions $\epsilon_t^\theta(\hat{\mathbf{x}}_t^i)$ is used as a measure of sample uncertainty.

To adapt AU to flow matching, we replace the score function with the learned velocity field $u_t^\theta(\mathbf{x}_t)$. Assuming an affine probability path, the clean data estimate $\hat{\mathbf{x}}_1$ can be recovered from the current state $\mathbf{x}_t$ and the predicted velocity as

$$\hat{\mathbf{x}}_1 = \frac{1}{\dot{\alpha}_t \beta_t - \dot{\beta}_t \alpha_t} \left( -\dot{\beta}_t \mathbf{x}_t + \beta_t u_t^\theta(\mathbf{x}_t) \right). \tag{35}$$

Starting from the estimated data $\hat{\mathbf{x}}_1$, we generate multiple perturbed states $\hat{\mathbf{x}}_t^i$ by applying the forward affine transformation. We then compute the element-wise variance of the velocity predictions $u_t^\theta(\hat{\mathbf{x}}_t^i)$, which serves as the aleatoric uncertainty estimate at time $t$. We aggregate the velocity uncertainties over the late-stage sampling steps to obtain a sample-level uncertainty estimate.

**BayesDiff (Kou et al., 2023).** In the diffusion-based formulation, BayesDiff estimates the uncertainty of the score function at each sampling step using Laplace Last Layer Approximation (LLLA) (Daxberger et al., 2021). The estimated score variance is then propagated through the diffusion dynamics to obtain uncertainty estimates for the generated samples.

To adapt BayesDiff to flow matching, we apply LLLA to the velocity field and estimate the variance of the predicted velocity at each time step. This velocity variance is subsequently propagated through the flow dynamics following the same uncertainty propagation scheme as in the original BayesDiff formulation.

**Generative Uncertainty (GenUnc) (Jazbec et al., 2025).** In the diffusion-based formulation, Generative Uncertainty (GenUnc) estimates sample uncertainty by sampling multiple model weights through LLLA and generating multiple images from the same noise realization. The resulting images are embedded into the CLIP (Radford et al., 2021) feature space, and the entropy of the extracted features is used as a sample-level uncertainty measure.

For flow matching, we follow the same procedure by sampling model weights, generating multiple samples from the same initial noise using the corresponding flow matching model, and computing the variance of the resulting CLIP features as the uncertainty estimate.

## F. Additional Results on Main Experiments

This section collects additional figures referenced by the main paper to complement the main experiments.

### F.1. Uncertainty-Based Filtering

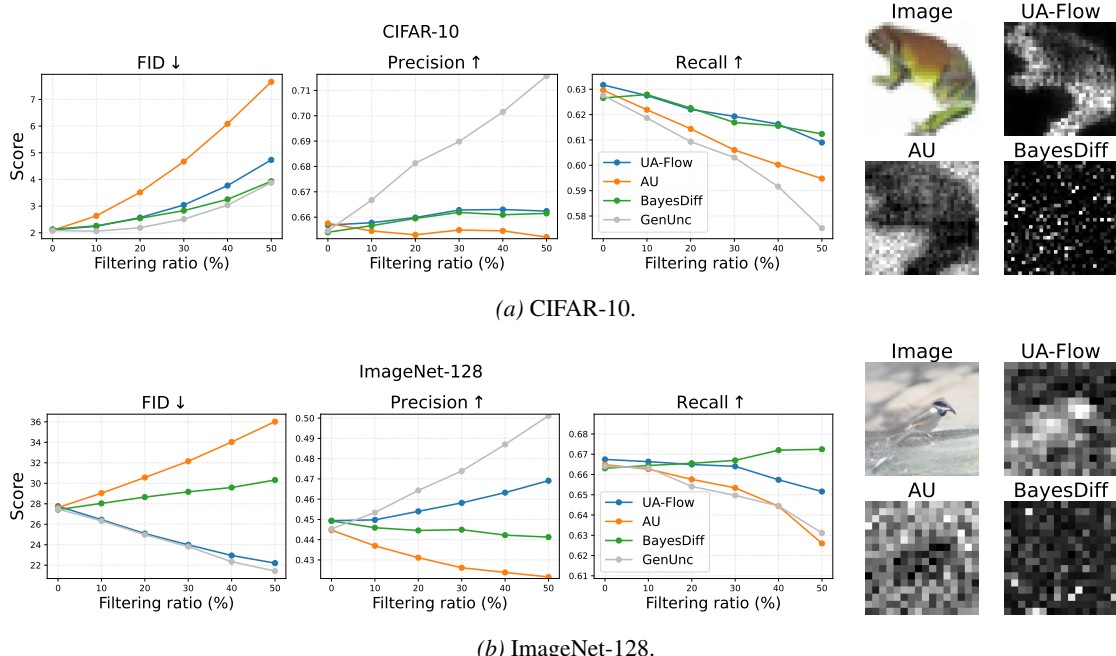

*(a)* CIFAR-10.

*(b)* ImageNet-128.

*Figure 6.* **Filtering high-uncertainty samples across datasets.** Left: generative quality metrics as a function of the filtering ratio. Right: example (latent) pixel-wise uncertainty maps produced by UA-Flow, AU, and BayesDiff for the same generated image. On CIFAR-10, uncertainty-based filtering primarily induces a precision-recall trade-off that can negatively affect FID despite high overall sample quality. On ImageNet-128, the trends are consistent with ImageNet-256: UA-Flow achieves lower FID and higher precision after filtering compared to AU and BayesDiff; GenUnc is included as a reference baseline.

We evaluate whether uncertainty provides a useful reliability signal by progressively filtering out high-uncertainty generated samples and tracking changes in FID and precision/recall. Figure 6 summarizes the results on CIFAR-10 and ImageNet-128. Across datasets, UA-Flow exhibits a consistent precision-recall trade-off under filtering: removing the most uncertain samples increases precision while reducing recall.

**CIFAR-10.** On CIFAR-10, the unfiltered sample quality is already high, so the precision gain from filtering can be offset by the loss in recall, resulting in flat or slightly worse FID as the filtering ratio increases.

**ImageNet-128.** On ImageNet-128, filtering more reliably improves fidelity: UA-Flow achieves lower FID and higher precision after filtering compared to AU and BayesDiff (with the expected decrease in recall). We include GenUnc as a reference baseline; it uses a domain-specific scalar uncertainty estimated in a CLIP embedding space, which is different in nature from element-wise uncertainty predicted by UA-Flow.

**Uncertainty maps.** Qualitatively, UA-Flow highlights spatially localized regions of high uncertainty, whereas AU often produces broadly inverted patterns and BayesDiff tends to yield noisier maps that do not clearly localize high-uncertainty regions.

### F.2. Uncertainty-Aware Guidance

F.2.1. UNCERTAINTY-AWARE CLASSIFIER GUIDANCE (U-CG)

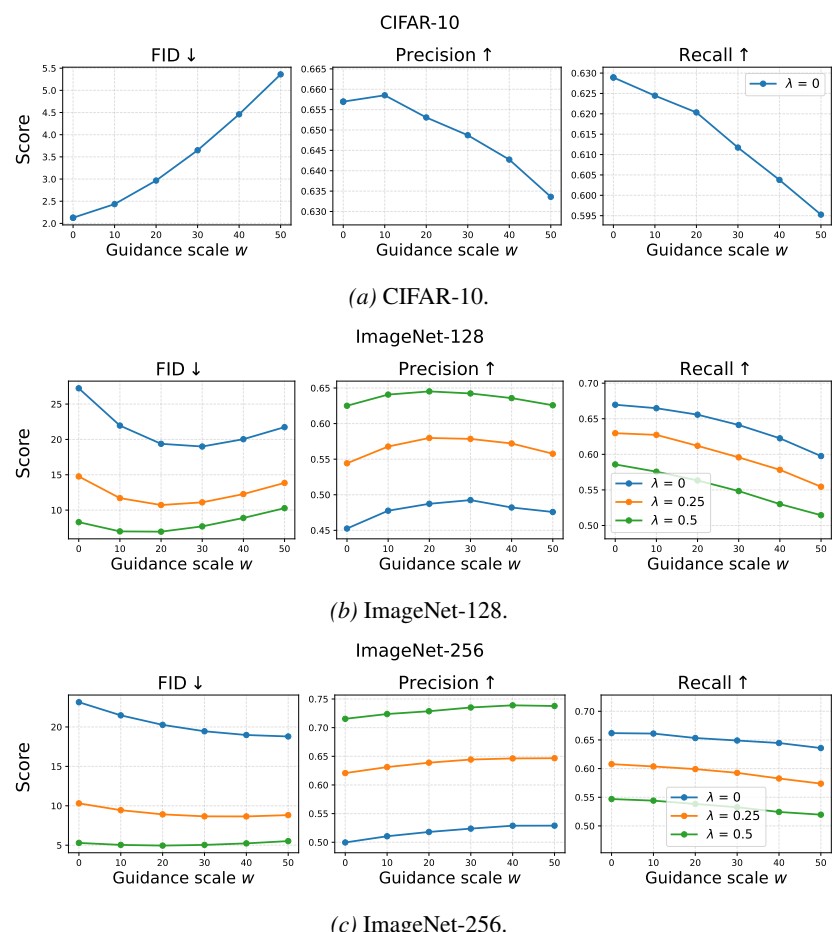

*(a)* CIFAR-10.

*(b)* ImageNet-128.

*(c)* ImageNet-256.

*Figure 7.* **Generation quality metrics as a function of the uncertainty-aware classifier guidance (U-CG) scale $w$ under fixed classifier-free guidance (CFG) scales.** Increasing $w$ induces a fidelity–diversity trade-off, improving precision while reducing recall. FID improves up to an optimal guidance strength, after which excessive guidance degrades performance.

We sweep the U-CG scale $w$ under fixed classifier-free guidance (CFG) scales. As shown in Figure 7, increasing $w$ induces a consistent precision-recall trade-off across datasets: precision typically increases or peaks at an intermediate $w$, while recall decreases as guidance becomes stronger. As a result, FID improves up to a dataset-dependent optimum, after which excessive guidance degrades performance. On CIFAR-10, where the baseline FID is already low, the same trade-off can translate into only marginal FID gains or a slight FID increase at larger $w$.

F.2.2. UNCERTAINTY-AWARE CLASSIFIER-FREE GUIDANCE (U-CFG)

We compare standard CFG having scale $\lambda$ with U-CFG, which uses a step-wise adaptive scale $\lambda^*$ clamped by $\lambda_{\max}$. In Figure 8, increasing the fixed CFG scale eventually degrades both precision and recall, leading to a sharp rise in FID at large $\lambda$. In contrast, U-CFG is substantially more robust as $\lambda_{\max}$ increases, with only mild changes in precision/recall and correspondingly smaller FID degradation. The violin plot shows that $\lambda^*$ is typically smaller in earlier sampling steps and larger in later steps, suggesting that U-CFG avoids over-guidance when the sample is still coarse and applies stronger guidance after the sample becomes more refined.

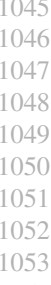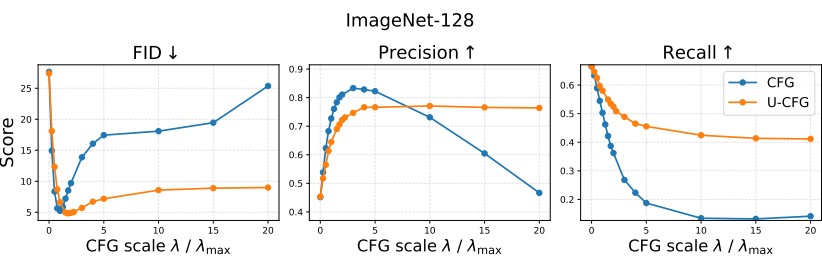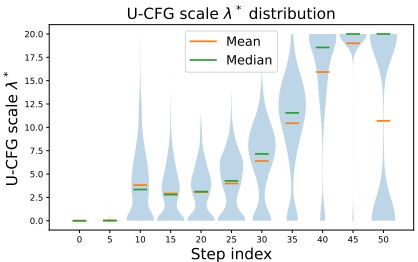

*(a)* FID, Precision, and Recall versus CFG scale $\lambda$ (CFG) or $\lambda_{\max}$ (U-CFG).

*(b)* Distribution of adaptive U-CFG scales across sampling steps (1,000 images).

*Figure 8.* (a) **FID, precision, and recall as a function of the fixed CFG scale $\lambda$ or the maximum scale $\lambda_{\max}$ of U-CFG on ImageNet-128.** CFG degrades sharply at large $\lambda$, while U-CFG remains more stable as $\lambda_{\max}$ increases. (b) **Violin plots of the adaptive U-CFG scale $\lambda^*$ across sampling steps 1,000 samples**. $\lambda^*$ tends to be smaller in early steps and larger in later steps.

## G. Supplementary Analyses

### G.1. Uncertainty Evolution in BayesDiff and UA-Flow

Figures 9 and 10 compare the evolution of uncertainty maps over the sampling trajectory for BayesDiff and UA-Flow. Each figure shows the generated image (top), the estimated velocity uncertainty at intermediate steps (middle), and the propagated state uncertainty at the corresponding steps (bottom). Notably, both methods produce velocity uncertainty maps with clear spatial correlation, indicating that uncertainty concentrates in specific regions rather than being uniformly distributed. More specifically, velocity uncertainty obtained from UA-Flow is typically less noisy.

Despite using the same variance propagation rule (Equation (5)), the propagated state uncertainty behaves very differently. For BayesDiff, the state uncertainty maps become largely unstructured and visually resemble noise, whereas UA-Flow yields state uncertainty maps that remain spatially coherent across steps. This behavior is consistent with Figure 11: BayesDiff exhibits a large uncertainty scale in the early sampling phase, and the corresponding early-step velocity uncertainty maps are also visibly noisy (e.g., step 12 in Figure 9). When such high-magnitude, noise-like velocity uncertainty is propagated, it can dominate the resulting state uncertainty and produce the unstructured, noise-like maps observed in Figure 9. In contrast, UA-Flow's learned heteroscedastic velocity uncertainty is more spatially coherent, leading to propagated state uncertainty maps that remain structured.

### G.2. Qualitative verification of uncertainty-based filtering

We next qualitatively assess whether UA-Flow's predicted uncertainty provides a meaningful reliability signal. To this end, we focus on two ImageNet-256 classes, `hen` and `teddy_bear`, and generate 1,000 samples per class using classifier-free guidance with scale $\lambda = 0.5$ to show more realistic images. For each generated image, we compute the same scalar uncertainty score used for filtering in Section 4.2, then respectively select the 25 images with the highest uncertainty and lowest uncertainty.

Figures 12 and 13 show clear qualitative separation between uncertainty extremes. High-uncertainty samples frequently exhibit severe structural distortions and cluttered scenes. In contrast, low-uncertainty samples tend to be sharply recognizable instances of the target class with clean textures and coherent global structure, indicating substantially higher perceptual fidelity.

At the same time, the low-uncertainty subsets also reveal a fidelity-diversity trade-off. For `hen`, low-uncertainty samples are dominated by canonical compositions, typically close-up head/torso views on simple backgrounds. For `teddy_bear`, low-uncertainty samples often depict centered plush toys under relatively uniform backgrounds. In contrast, the high-uncertainty subsets span more diverse contexts, poses, and compositions, albeit with noticeably lower fidelity. Overall, these results provide qualitative evidence that UA-Flow uncertainty is aligned with sample-level quality, and that uncertainty-based filtering behaves as observed in Section 4.2: retaining low-uncertainty (high-fidelity) samples while implicitly reducing diversity.

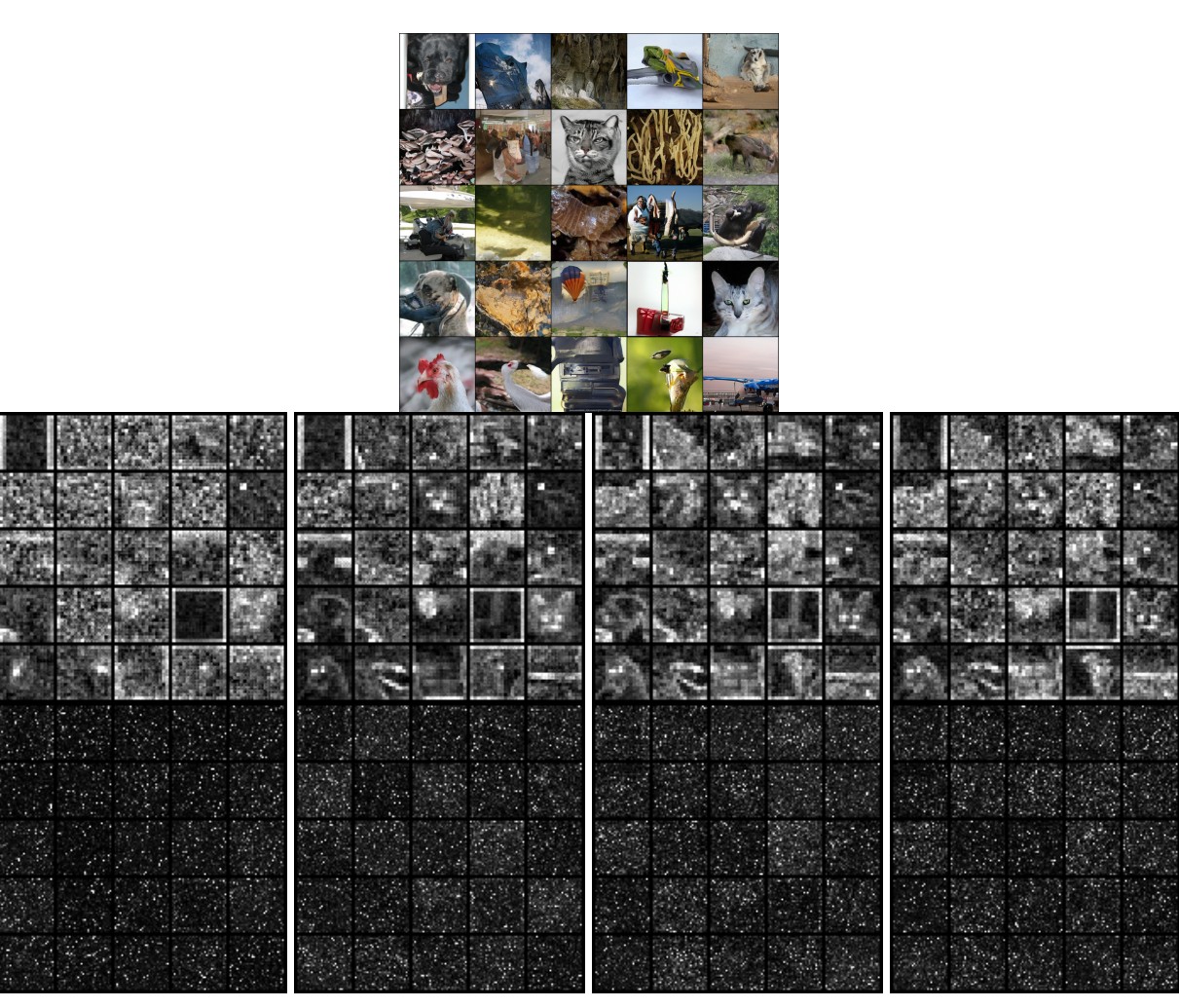

*Figure 9.* **Qualitative uncertainty evolution for BayesDiff.** Images are sampled on ImageNet-256 without guidance (no classifier guidance or classifier-free guidance). *Top*: the generated sample. *Middle*: BayesDiff's velocity uncertainty maps at intermediate sampling steps 12/24/36/48 (left to right). *Bottom*: state uncertainty maps obtained by propagating the velocity uncertainty through the sampling dynamics using the same variance propagation rule as UA-Flow. While the velocity uncertainty exhibits spatial correlation, the propagated state uncertainty becomes largely noise-like and fails to preserve coherent spatial structure.

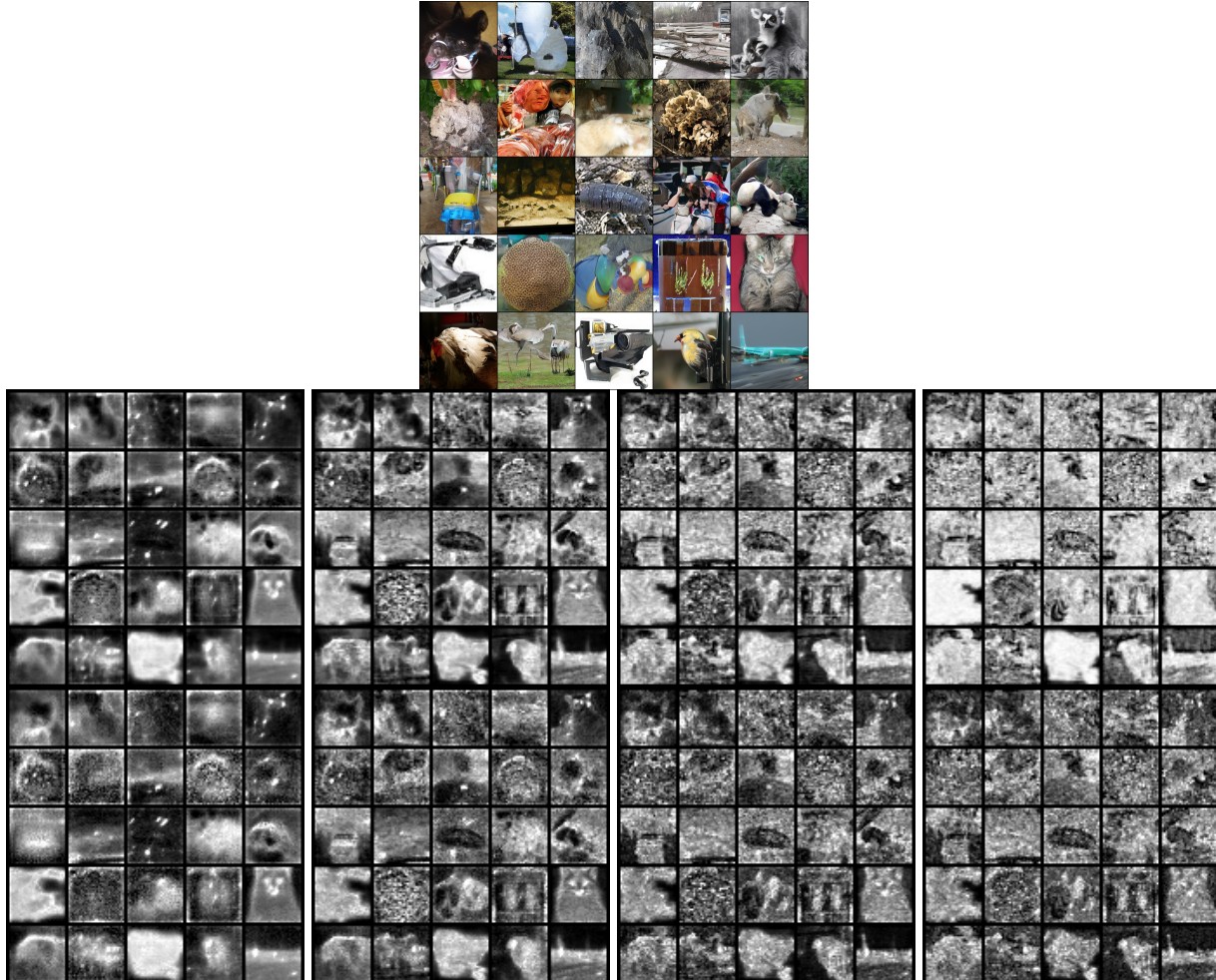

*Figure 10.* **Qualitative uncertainty evolution for UA-Flow.** Images are sampled on ImageNet-256 without guidance (no classifier guidance or classifier-free guidance). *Top*: the generated sample. *Middle*: UA-Flow's velocity uncertainty maps at intermediate sampling steps 12/24/36/48 (left to right). *Bottom*: propagated state uncertainty maps computed from the velocity uncertainty via variance propagation. In contrast to BayesDiff, the state uncertainty remains spatially coherent and tracks structured regions throughout sampling.

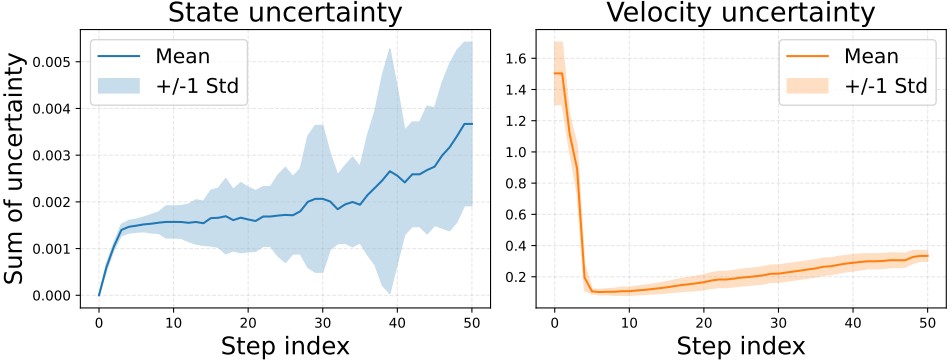

*Figure 11.* **Uncertainty scale over sampling steps for BayesDiff.** We quantify the magnitude of uncertainty at each sampling step by summing the uncertainty map over all elements (spatial locations and channels) for each sample. The plot reports the mean ± standard deviation of this scalar summary across the $5 \times 5$ sample grid shown in Figure 9. The noisy state uncertainty maps in Figure 9 are consistent with the elevated uncertainty scale in the early sampling phase, suggesting that BayesDiff's uncertainty is poorly calibrated across sampling time and can dominate variance propagation.

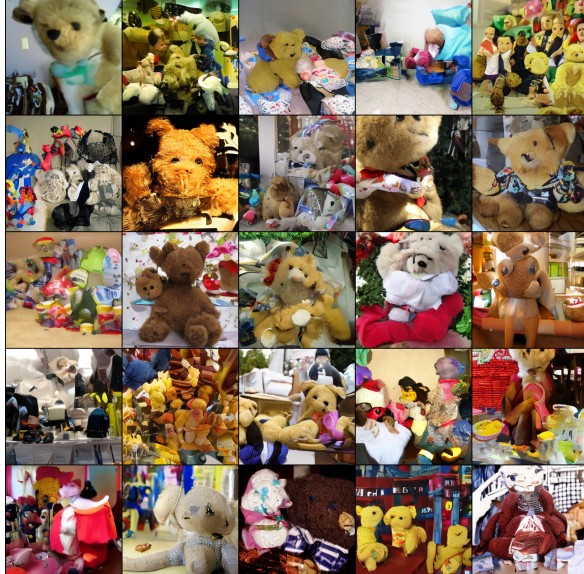

*Figure 12.* **Teddy bear (ImageNet-256, CFG** $\lambda = 0.5$**): lowest vs. highest uncertainty samples.** We generate 1,000 samples and select the 25 lowest-uncertainty images (top) and the 25 highest-uncertainty images (bottom) using the same uncertainty aggregation and ranking procedure as in the main paper. Low-uncertainty samples are visually clean and class-consistent, while high-uncertainty samples often contain clutter, distortions, or weak class identity.

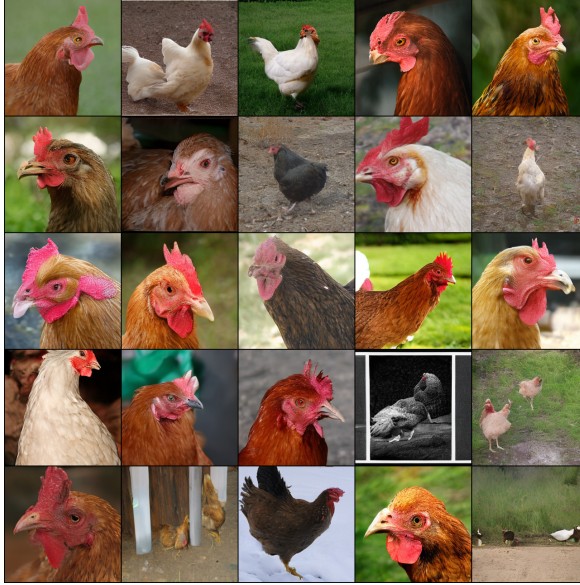
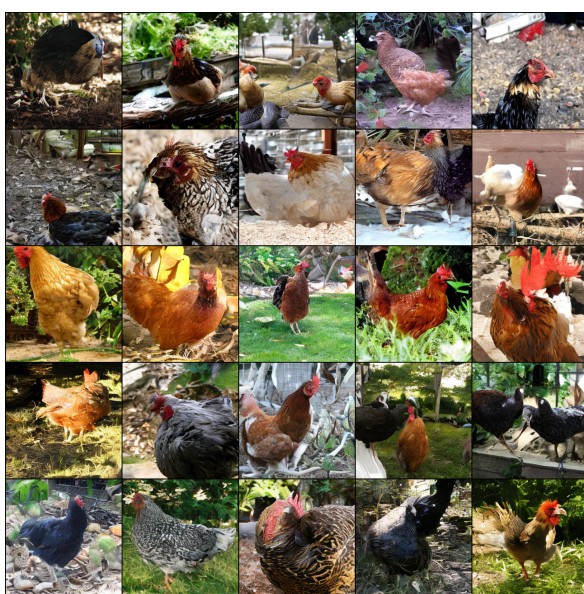

*Figure 13.* **Hen (ImageNet-256, CFG** $\lambda = 0.5$**): lowest vs. highest uncertainty samples.** We generate 1,000 samples and select the 25 lowest-uncertainty images (top) and the 25 highest-uncertainty images (bottom). Low uncertainty corresponds to high-fidelity, easily recognizable hens (often in canonical views), whereas high uncertainty corresponds to less reliable generations with more frequent artifacts or reduced class salience, illustrating a qualitative fidelity–diversity trade-off.

### G.3. Ablations on Uncertainty Estimation

G.3.1. EFFECT OF THE VARIANCE TERM $\mathrm{Var}(\bar{u}_t^\theta(\mathbf{x}_t))$ IN EQUATION (23)

In Equation (23), the velocity variance can be decomposed into two terms: the predicted heteroscedastic term $(\sigma_t^\theta(\bar{\mathbf{x}}_t))^2$ and the additional variance induced by the spread of $\mathbf{x}_t$, $\mathrm{Var}(\bar{u}_t^\theta(\mathbf{x}_t))$. In the main text, we drop the latter to avoid extra Monte Carlo (MC) computation. Here, we evaluate its practical impact by explicitly estimating $\mathrm{Var}(\bar{u}_t^\theta(\mathbf{x}_t))$ with MC sampling.

Specifically, for each step $t$ we draw $K = 10$ samples $\mathbf{x}_{t,i} \sim \mathcal{N}(\bar{\mathbf{x}}_t, \mathrm{Var}[\mathbf{x}_t])$ and compute the empirical (diagonal) variance of $\bar{u}_t^\theta(\mathbf{x}_{t,i})$ across $i \in \{1, \ldots, K\}$. We then define the *total* velocity variance as $(\sigma_t^\theta(\bar{\mathbf{x}}_t))^2 + \mathrm{Var}(\bar{u}_t^\theta(\mathbf{x}_t))$ and repeat the same uncertainty-based filtering protocol as in Section 4.2 (ImageNet-256, UA-Flow, no guidance).

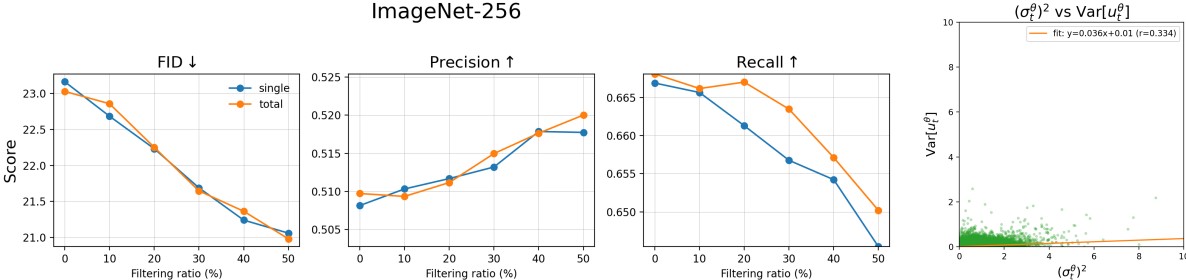

*Figure 14.* (a) **Effect of explicitly including** $\mathrm{Var}(\bar{u}_t^\theta(\mathbf{x}_t))$ **in velocity variance for uncertainty-based filtering (ImageNet-256, UA-Flow, no guidance).** *Single* uses only $(\sigma_t^\theta(\bar{\mathbf{x}}_t))^2$ as in the main text, while *total* adds the MC-estimated term $\mathrm{Var}(\bar{u}_t^\theta(\mathbf{x}_t))$ with $K = 10$ samples. Across filtering ratios, FID/precision/recall curves remain largely unchanged. (b) **Comparing the two terms in Equation (23):** **Scatter plot of** $\mathrm{Var}(\bar{u}_t^\theta(\mathbf{x}_t))$ **(MC estimate with $K = 10$) versus** $(\sigma_t^\theta(\bar{\mathbf{x}}_t))^2$. The heteroscedastic term dominates, consistent with the negligible impact observed in filtering.

As shown in Figure 14, explicitly incorporating $\mathrm{Var}(\bar{u}_t^\theta(\mathbf{x}_t))$ does not significantly change the filtering behavior. Since estimating this term requires $K$ additional forward evaluations of $\bar{u}_t^\theta$ per sampling step, we omit it in practice as it does not provide a clear benefit relative to its computational overhead.

Finally, we analyze the relative magnitude of the two variance terms. Figure 14 illustrates scatter plots of $10^6$ randomly the MC-estimated $\mathrm{Var}(\bar{u}_t^\theta(\mathbf{x}_t))$ against $(\sigma_t^\theta(\bar{\mathbf{x}}_t))^2$ over sampling steps and spatial locations through generating 1,000 images. We observe a positive correlation, but the estimated slope is small: a linear fit yields $\mathrm{Var}(\bar{u}_t^\theta(\mathbf{x}_t)) \approx 0.036 \, (\sigma_t^\theta(\bar{\mathbf{x}}_t))^2 + 0.01$ with Pearson correlation $r = 0.334$. This indicates that the heteroscedastic term is dominant, and the MC-estimated contribution typically adds only a small ($\sim 3.6\%$) variance gain.

G.3.2. EFFECT OF THE NUMBER OF HUTCHINSON PROBES $S$

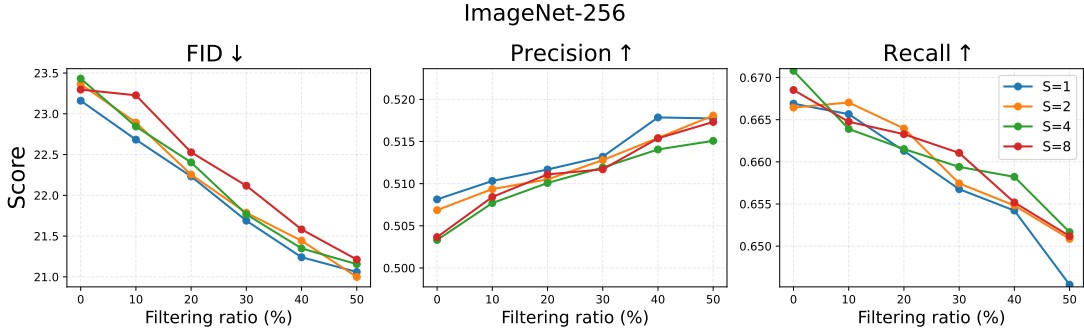

*Figure 15.* **Effect of the number of Hutchinson probes $S$ on uncertainty-based filtering (ImageNet-256, UA-Flow, no guidance).** Generation metrics are plotted as a function of the filtering ratio for $S \in \{1, 2, 4, 8\}$.

In UA-Flow, the state variance update (Equation (5)) includes a covariance term approximated by the Hutchinson's diagonal estimator as Equation (7). Following the main filtering protocol in Section 4.2 (ImageNet-256, UA-Flow, no guidance), we evaluate the sensitivity of uncertainty-based filtering to the number of probes $S$ used in the diagonal estimator. As shown in

Figure 15, sweeping $S \in \{1, 2, 4, 8\}$ does not largely change curves for FID, precision, and recall across filtering ratios. This indicates that increasing $S$ does not largely change the resulting uncertainty of samples in this setting. Therefore, we use $S = 1$ by default as it provides a good accuracy without extra computational cost.

### G.3.3. EFFECT OF COVARIANCE APPROXIMATION

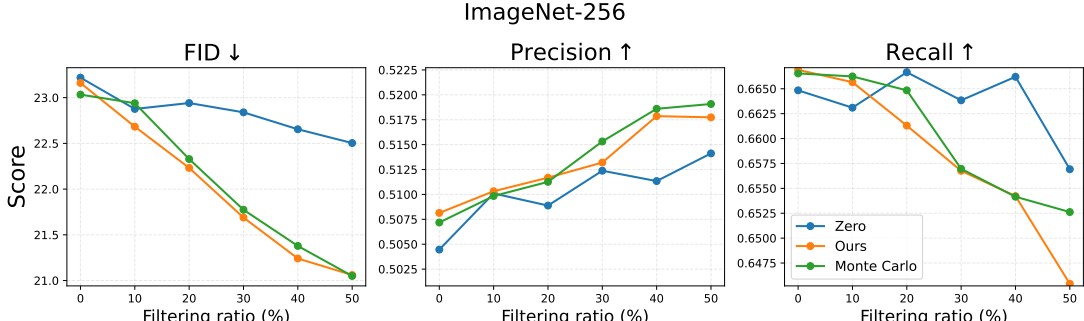

*Figure 16.* **Effect of covariance approximation on uncertainty-based filtering (ImageNet-256, UA-Flow, no guidance).** We compare Option 1 (zero covariance), Option 2 (ours; JVP-based), and Option 3 (Monte Carlo) by plotting generation metrics versus the filtering ratio.

We then compare different covariance approximations in Equation (29) for the variance propagation step while keeping the rest of the filtering pipeline fixed (ImageNet-256, UA-Flow, no guidance). Figure 16 compares: (i) *Option 1 (zero)*, which drops the covariance term; (ii) *Option 2 (ours)*, which uses the proposed JVP-based approximation with $S = 1$; and (iii) *Option 3 (Monte Carlo)*, which uses a Monte Carlo estimator and incurs higher computational cost with 10 Monte Carlo samples used as default in BayesDiff (Kou et al., 2023), i.e. $S = 10$. Dropping the covariance term yields the worst filtering behavior, with the highest FID and a substantially weaker precision-recall trade-off. In contrast, Option 2 and Option 3 yield similar performance across filtering ratios, indicating that the proposed approximation captures the essential covariance structure at a fraction of the computational cost of Monte Carlo estimation. Overall, these results highlight both the necessity of accounting for covariance in accurate uncertainty estimation and the practical benefits of Option 2.

### G.4. Empirical validation of conditional–unconditional uncertainty correlation for U-CFG

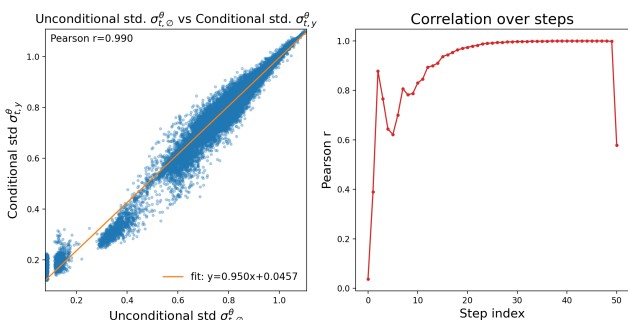

*Figure 17.* **Correlation between unconditional and conditional velocity uncertainty.** *Left:* Scatter plot of unconditional vs. conditional predicted standard deviations $(\sigma_{t,\emptyset}^{\theta}, \sigma_{t,y}^{\theta})$, using $10^6$ randomly sampled elements across all sampling steps and 1,000 generated samples (no CFG, $\lambda = 0$). *Right:* Pearson correlation between $\sigma_{t,\emptyset}^{\theta}$ and $\sigma_{t,y}^{\theta}$ computed at each sampling step (aggregated over all elements and samples).

Uncertainty-aware classifier-free guidance (U-CFG) combines conditional and unconditional predictions and approximates the variance of the extrapolated velocity under the assumption that the conditional and unconditional uncertainties are strongly correlated (see Equation (10)). To validate this assumption empirically, we measure the relationship between the predicted conditional and unconditional velocity standard deviations along the sampling trajectory.

We generate 1,000 samples without CFG (i.e. $\lambda = 0$) using the same sampling configuration as in our main experiments. At each intermediate sampling step, we evaluate the model twice at the current state: once with the class condition $y$ and once

with the null condition $\varnothing$, obtaining element-wise standard deviations $\sigma^{\theta}_{t,y}(\bar{\mathbf{x}}_t)$ and $\sigma^{\theta}_{t,\varnothing}(\bar{\mathbf{x}}_t)$, respectively. To visualize the global relationship across time and spatial locations, we randomly subsample $10^6$ element pairs $\left(\sigma^{\theta}_{t,\varnothing}, \sigma^{\theta}_{t,y}\right)$ from all steps and all samples.

Figure 17 (left) shows an almost linear relationship between $\sigma^{\theta}_{t,\emptyset}$ and $\sigma^{\theta}_{t,y}$, with Pearson correlation $r \approx 0.99$ over the subsampled elements. Figure 17 (right) reports the per-step Pearson correlation computed over all elements, showing that the correlation is close to $1.0$ for most steps, with deviations limited to a few early/late steps. Overall, these results provide empirical support for treating the conditional and unconditional uncertainty predictions as strongly correlated when defining the U-CFG uncertainty used in the main paper.

## H. Samples and Uncertainties

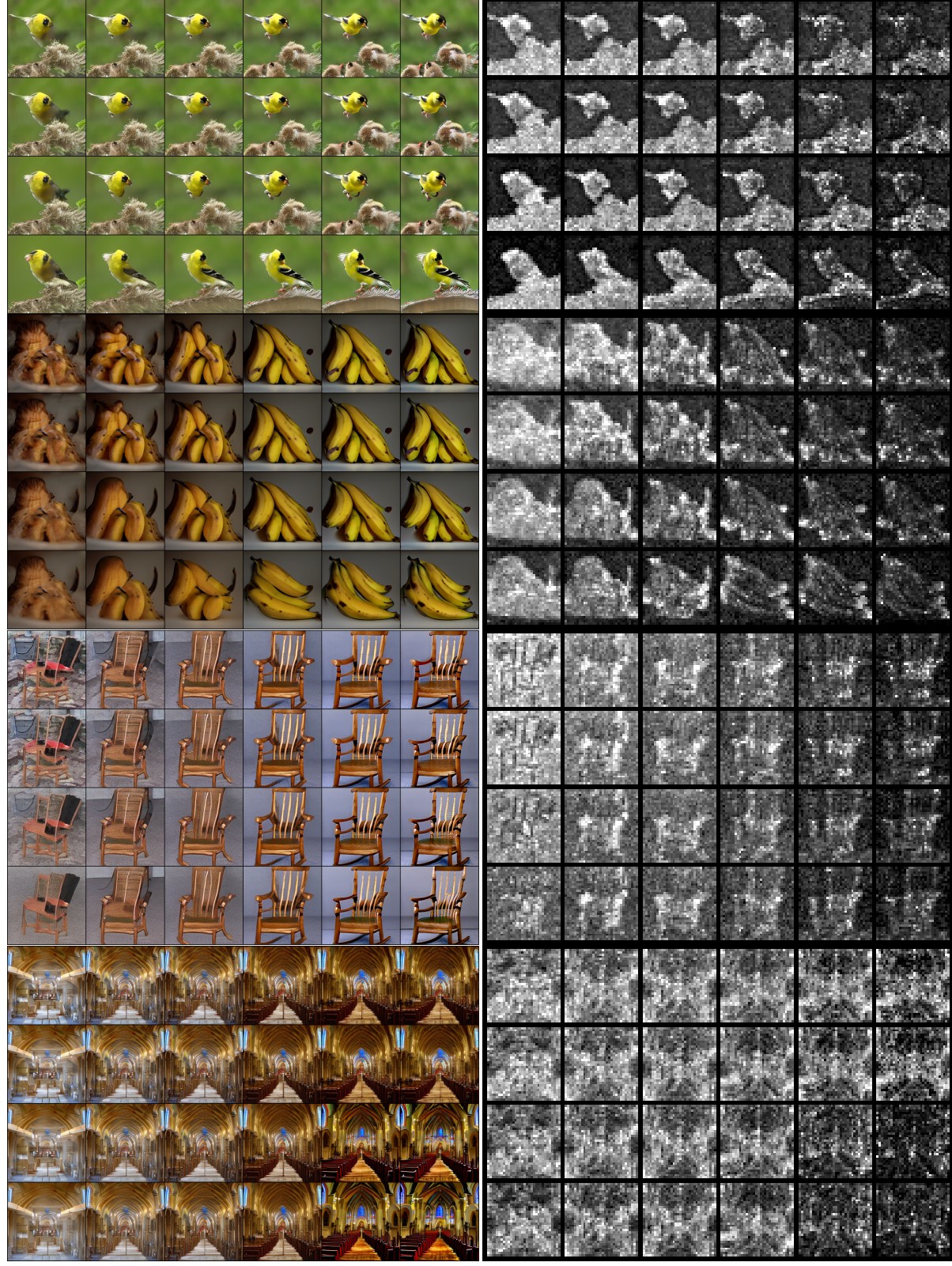

*Figure 18.* **Selected ImageNet-256 sample grids under uncertainty-aware guidance sweeps.** Rows sweep the U-CG scale $w \in \{0, 10, 30, 50\}$ and columns sweep the maximum U-CFG scale $\lambda_{\max} \in \{0, 1, 2, 5, 10, 20\}$. Left: generated samples. Right: predicted latent pixel-wise uncertainty maps.

