# OpenReview forum: "Flow Matching with Uncertainty Quantification and Guidance"
_ICML.cc/2026/Conference — Submitted to ICML 2026_

### Official Review · Reviewer_9dW2 · 2026-02-28

**Soundness:** 3
**Presentation:** 2
**Significance:** 2
**Originality:** 1
**Overall Recommendation:** 3
**Confidence:** 3

**Summary:**

The paper proposes UA-Flow, an extension of flow matching that predicts the velocity field together with the heteroscedastic uncertainty (variance). It estimates the per-sample uncertainity by propagating vector field uncertainty through the ODE dynamics. It then uses this uncertainty in two guidance schemes: uncertainty-aware classifier guidance and classifier-free guidance. Experiments on image generation show that UA-Flow improves generation quality by filtering our those highly uncertain samples.

**Compliance With Llm Reviewing Policy:**

Affirmed.

**Final Justification:**

I still have concerns about the effectiveness of the proposed method, since the performance gain compared to the baseline methods is very limited. Moreover, the effectiveness of the method in high-dimensional settings remains unclear.

**Key Questions For Authors:**

- Given more computational costs, can UA-Flow outperform GenUnc?
- In the CIFAR case, the performance gets worse if we remove 10% bad samples. What if we remove only 1%?
- Following the previous question, the explanation of worse performance is that "the unfiltered sample quality is already high". How can we quantify the sample quality in a specific application?

**Limitations:**

See weakness

**Strengths And Weaknesses:**

Strength:
- The paper is well-written.
- Simple modification. Adding the variance and training with an NLL style loss is straightforward.
- Uncertainty-guided sampling is an interesting direction; U-CFG is simple to implement.

Weakness:
- Limited comceptual novelty. The high-level idea of this paper is using generative uncertainty to filter bad samples. The idea is the same as that in, e.g., GenUnc, alghough there are some differences in terms of the type of model, uncertainty definition.
- A practical concern is that performance is worse than GenUnc. Even though GenUnc is more computational heavy as in Table 1, it is unclear if with more computational costs, the method can outperform GenUnc.
- Another concern is that in CIFAR10 the the performance gets worse if we use UA-Flow to filter out samples. It remains unclear if UA-Flow is generally useful.
- The evaluation is limited to image generation.

---

> ### Author Rebuttal · Authors · 2026-03-31
>
> We thank the reviewer for the detailed questions.
>
> ### [W2/Q1] Performance worse than GenUnc; can UA-Flow outperform with more compute?
>
> Yes. Since UA-Flow predicts mean and variance at the endpoint, we draw K=6 latent samples without additional flow-model passes, decode through the VAE, and compute CLIP-embedding variance, replicating GenUnc's signal. UA-Flow+CLIP **outperforms** GenUnc in FID at every filtering ratio (e.g., **16.224 vs 17.408** at 50%) while requiring only 12.3 TFLOPs (one flow pass + VAE/CLIP post-processing) versus GenUnc's 33.7 TFLOPs (multiple weight-perturbed flow passes).
>
> Please see our response to Reviewer EtRa [W2] for full results.
>
> ### [W1] Limited conceptual novelty vs. GenUnc.
>
> First, the methodological contribution of UA-Flow, linearized variance propagation through the flow ODE with a trained variance head, is architecturally distinct from GenUnc's weight-perturbation approach.
>
> Second, The practical contributions are not only uncertainty quantification (UQ) of flow-based model to remove degraded samples, but also direct application of the uncertainty during generation process. GenUnc produces a scalar quality score via CLIP embeddings, which is domain-specific and applicable only to filtering. However UA-Flow produces spatially-resolved uncertainty from the velocity field, enabling three distinct practical capabilities: (1) filtering, (2) U-CG guidance, and (3) U-CFG guidance. The latter two are impossible with GenUnc's post-hoc scalar signal.
>
> ### [W3/Q2] CIFAR-10 performance worse after filtering; what about 1%?
>
> While FID increases in this case, filtering has the effect of trading diversity for realism as indicated by improving precision and decreasing recall. Moreover, combining UA-Flow with CLIP embedding variance yields FID improvements even on CIFAR-10 (see below).
>
> We run fine-grained CIFAR-10 filtering at {0%, 1%, 2%, 5%} with 3 random seeds to select 50k samples among 100k samples for the evalutations. At 1–2% removal, all metrics stay within noise of the baseline, and none deviate by more than one standard deviation. Furthermore, combining UA-Flow with CLIP embedding variance (as in our response to [W2/Q1]) yields FID improvements on CIFAR-10 as well. At 1–2% removal, UA-Flow+CLIP improves FID beyond one standard deviation (2.132±.008 vs 2.167±.022)
>
> **Filtering experiment on CIFAR-10 (UA-Flow)**
>
> |Filter %|FID↓|Precision↑|Recall↑|
> |---|---|---|---|
> |0%|**2.111±.009**|.6564±.0026|.6250±.0008|
> |1%|2.117±.013|**.6589±.0003**|.6250±.0003|
> |2%|**2.111±.011**|.6586±.0015|**.6253±.0016**|
> |5%|2.144±.015|.6578±.0017|.6235±.0006|
>
> **Filtering experiment on CIFAR-10 (UA-Flow + CLIP)**
>
> |Filter %|FID↓|Precision↑|Recall↑|
> |---|---|---|---|
> |0%|2.167±.022|.6587±.0004|**.6314±.0012**|
> |1%|2.133±.014|.6591±.0005|.6293±.0018|
> |2%|**2.132±.008**|**.6621±.0018**|.6295±.0013|
> |5%|2.171±.016|.6630±.0021|.6246±.0021|
>
> ### [Q3] The explanation of worse performance is that "the unfiltered sample quality is already high". How to quantify sample quality in a specific application?
>
> The CIFAR-10 model already achieves FID ~2.1 near the dataset optimum, so the precision-recall tradeoff from filtering degrades FID only at 5% removal, in contrast to ImageNet where clear FID gains appear immediately. The utility of uncertainty-based filtering scales with generation difficulty.
>
> ### [W4] Only image experiments. (common response to Reviewer bhpF [W4/Q4] and eA44 [W1])
>
> We conduct two additional experiments to demonstrate UA-Flow beyond image generation. Per-sample uncertainty is obtained by summing element-wise variance, providing a scalar ranking signal.
>
> **(A) 2D checkerboard**: We train a flow matching MLP with a variance head on a 2D checkerboard distribution and generate 10,000 samples. The variance head assigns 8× higher mean uncertainty outside the board than inside (7.08e-4 vs 8.79e-5). Filtering by predicted uncertainty monotonically improves coverage (fraction of retained samples lying inside the checkerboard):
>
> |Filter %|Coverage (%)↑|
> |---|---|
> |0%|95.7|
> |1%|96.7|
> |2.5%|97.9|
> |5%|98.8|
> |10%|**99.4**|
>
> **(B) Time-series sine wave**: A 1D UNet with variance head generates sine wave continuations conditioned on previous context (100 inputs x 100 samples each). Filtering by predicted uncertainty monotonically reduces reconstruction error:
>
> |Filter %|Mean MSE ($\times 10^{-5}$)↓|
> |---|---|
> |0%|146.5|
> |10%|38.73|
> |25%|21.10|
> |50%|17.65|
> |90%|**7.795**|
>
> The linked figures confirms that high-uncertainty samples are generally degraded, while low-uncertainty samples remain accurate after filtering:
> https://i.postimg.cc/tCnt8ZKn/checkerboard.png and https://i.postimg.cc/WzFgRCHv/timeseries.png .
>
> These results span two non-image domains (spatial 2D, temporal 1D) and two architectures (MLP, 1D UNet).

---

> > ### Author Rebuttal · Reviewer_9dW2 · 2026-04-02
> >
> > I thank the authors for the answers to my questions and provide additional experiments. However, I still have the concern about the effectiveness of the method. Although the authors find that their framework in combination with CLIP features can  outperform GenUn in Imagenet, the performance gain is relatively modest. In the CIFAR case, the performance is even worse after filtering. The authors claimed that "filtering has the effect of trading diversity for realism as indicated by improving precision and decreasing recall", but the precision improvement is very small. Moreover, the additional experiments are 2D examples and simple sine wave, which cannot illustrate the effectiveness of the method in high-dimensional applications.
> >
> > Based on the above, I will maintain the score.

---

> > > ### Author Response · Authors · 2026-04-03
> > >
> > > We thank the reviewer for the continued engagement. We address the remaining concerns with new evidence.
> > >
> > > ### 1. modest filtering performance of UA-Flow on ImageNet-256 compared to GenUnc.
> > >
> > > Even though UA-Flow outperform GenUnc within a close range (17.408vs16.224), UA-Flow is **2.7x faster** than GenUnc. **The lower computational cost and better FID score on ImageNet** would show the better applicability than GenUnc for the filtering task. Moreover, UA-flow has its additional novelty of its capability of uncertainty-aware guidance, not explored in GenUnc as stated in the response to the [W1].
> > >
> > > ### 2. Extended CIFAR-10 Filtering (0–50%)
> > >
> > > To relieve the reviewer's concern about small preicision improvement of UA-Flow + CLIP on the CIFAR-10, we extend the filtering rate from 10% to 50%. UA-Flow + CLIP achieves 0.430 precision gain at 50% removal. The results shows **the more significant tradeoff** with the combination of CLIP.
> > >
> > > **The FID increase does not indicate method failure.** it is the **expected** consequence of trading diversity for realism on a near-optimal baseline (FID around 2.1), as discussed in our [Q3] response. FID conflates precision and recall [1], so it worsens on CIFAR-10 when diverse but lower-fidelity samples are removed even as precision genuinely improves. On ImageNet-256, where baseline FID ( around 23) leaves headroom, UA-Flow + CLIP achieves clear FID gains (23.4 → 16.2 at 50%) while outperforming GenUnc at every ratio.
> > >
> > > **UA-Flow + CLIP**:
> > >
> > > |Filter %|FID ↓|Precision ↑|Recall ↑|
> > > |-|-|-|-|
> > > |0%|2.170|0.6588|0.6316|
> > > |10%|2.299|0.6680|0.6198|
> > > |20%|2.622|0.6774|0.6107|
> > > |30%|3.216|0.6843|0.6060|
> > > |40%|4.036|0.6917|0.5950|
> > > |50%|5.376|**0.7018**|0.5816|
> > >
> > > * [1] T Kynkäänniemi et al., Improved precision and recall metric for assessing generative models, NeurIPS 2019.
> > >
> > > ### 3. PushT Robot Policy Experiment (Additional Experiment on Non-image Domain)
> > >
> > > To address the concern that non-image experiments are limited to simple domains, we evaluate UA-Flow on **PushT** [2], a robot manipulation benchmark where a robot policy must push a T-shaped block to a target pose. The image and robot state are conditional inputs to UA-Flow. At each step, we generate 20 candidate action chunks, filter out the 10% with the highest uncertainty, and sample one action chunk from the remainder, with uncertainty aggregated by summing across all action dimensions (following our [W4] response). Results are over 200 rollouts, and maximum reward is 1.0. A rollout is considered successful when reward exceeds 0.95. we report the 1st and 5th percentiles of the reward (q1% and q5%, respectively) to **capture worst-case performance**, in addition to success rate, mean reward and predicted uncertainty.
> > >
> > > UA-Flow filtering at 10% improves **all metrics**. **This demonstrates UA-Flow's utility beyond image generation and simple 2D checkboard and time series sine wave tasks.** In particular, we believe that this shows UA-Flow can meaningfully reduce rare but potentially dangerous failure cases in robotics.
> > >
> > > |Filter %|Success Rate ↑|Mean Reward ↑|q1% ↑|q5% ↑|Mean Unc. ($\times10^{-4}$) ↓|
> > > |-|-|-|-|-|-|
> > > |0%|0.5950|0.8702|0.3148|0.5591|5.15|
> > > |10%|**0.7350**|**0.9053**|**0.4496**|**0.5691**|**4.88**|
> > >
> > > * [2] F. Zhang and M. Gienger. Affordance-based Robot Manipulation with Flow Matching. arXiv, 2024.

---

### Official Review · Reviewer_eA44 · 2026-03-10

**Soundness:** 3
**Presentation:** 4
**Significance:** 3
**Originality:** 3
**Overall Recommendation:** 4
**Confidence:** 2

**Summary:**

The paper proposes an uncertainty-aware flow matching (UC-Flow), which models uncertainty in the velocity field and propagates it through the ODE dynamics. Based on this uncertainty estimate, the author further proposes two guidance strategies (U-CG, U-CFG). Experiments on CIFAR-10 and ImageNet show that the estimated uncertainty can be used for sample filtering and uncertainty-aware classifier guidance, leading to improved FID.

**Compliance With Llm Reviewing Policy:**

Affirmed.

**Final Justification:**

The paper presents a reasonably well-approached approach to incorporating uncertainty into flow matching, with clear empirical benefits for filtering and guidance. While the novelty and evaluation scope still feel somewhat limited, I think the contribution is meaningful and potentially useful. The rebuttal addressed my concerns, particularly on the reliability side of the uncertainty estimates, and reinforced my concerns, so I maintain my score.

**Key Questions For Authors:**

1. The proposed uncertainty estimation seems to work well for filtering and guidance. However, I was less clear on how strongly the current results support the uncertainty scores as a measure of sample reliability, beyond their practical usefulness. I had a similar question when reading the ablations on uncertainty estimation, which are mainly evaluated through filtering.

2. Since the method is presented as an uncertainty estimation approach for flow matching, I was curious how sensitive it is to the choice of architecture (FM variants). Would similar improvements in filtering and guidance remain similar? if the architecture changes but the dataset is fixed?

**Limitations:**

This paper includes an impact statement, but the limitations of the proposed method are not discussed in detail. A brief discussion of these limitations would strengthen the paper.

**Strengths And Weaknesses:**

**Soundness**
* The paper presents a coherent framework, and the experiments support that the proposed uncertainty can be useful for filtering and guidance.
* Although I was not able to fully understand every approximation in the uncertainty modeling and propagation, the overall approach seems sensible and reasonably well supported by the empirical results.
* However, evidence for its usefulness in filtering and guidance is clearer than the evidence for how directly the uncertainty scores reflect sample reliability.

**Presentation**
* The paper is generally clear and easy to follow.
* The motivation, method, and experiments are presented in a fairly organized way, and I did not have difficulty understanding the main ideas at a high level.

**Significance**
* I think the problem is relevant, since uncertainty estimation for flow matching seems worth exploring and could be useful in practice.
* The results are encouraging, especially on ImageNet, although the overall impact feels somewhat limited by the evaluation scope.

**Originality**
* The novelty feels moderate but meaningful.
* The paper brings uncertainty modeling and guidance into the flow matching setting in a fairly reasonable way.

---

> ### Author Rebuttal · Authors · 2026-03-30
>
> We thank the reviewer for the positive assessment and thoughtful questions.
>
> ### [Q1] Uncertainty as a measure of sample reliability beyond filtering.
>
> We conduct indirect calibration analyses as a way to show the capability of UA-Flow's uncertainty as a measure of sample reliability.
> we obtain the latent of ImageNet validation images,z_1,  noise it to z_t and compare the predicted latent \hat{z}_t against the true latent. We evaluate state uncertainty $\mathrm{Var}[z_1 \mid z_t]$ starting from the noised latent against ground-truth reconstruction error using PIT ECE and Brier score, comparing UA-Flow, BayesDiff, and AU. UA-Flow is better calibrated than both baselines even before post-hoc calibration, and retains its advantage after calibration (**PIT ECE 0.0008 / Brier 0.0481** at t=0.9 vs. 0.0038/0.0505 for BayesDiff and 0.0012/0.0496 for AU). This provides evidence that the uncertainty scores are a meaningful measure of sample reliability beyond their practical usefulness for filtering.
>
> Please see our response to Reviewer bhpF [W2/Q2] for full details.
>
> ### [W1] (Significance) the overall impact feels somewhat limited by the evaluation scope
>
> We extend our uncertainty evaluation beyond non-image domains, 2D checkerboard and sine wave time-series dataset, and show that uncertainty-based filtering consistently removes degraded samples across both experiments. We use an MLP for the 2D checkerboard and a 1D UNet for the sine wave time-series.
>
> Please see our response to Reviewer 9dW2 [W4] for full results.
>
> ### [Q2] Sensitivity to FM architecture variants
>
> UA-Flow's uncertainty depends only on the velocity field $u_\theta$ and its Jacobian, which is architecture-agnostic properties of any flow matching model. The variance head is a lightweight add-on (same input/output dimensions as the velocity head) attachable to any backbone, so any architecture that performs well in vanilla FM would work equally well with UA-Flow.
>
> Our experiments already span four architectures: ADM-based convolutional model (CIFAR-10), DiT-B/2 transformer (ImageNet). Additionally, we train UA-Flow with an MLP (2D checkerboard, [W1]) and a 1D UNet (sine wave time-series, [W1]). Consistent uncertainty quantification across all four supports architectural robustness.

---

> > ### Author Rebuttal · Reviewer_eA44 · 2026-04-02
> >
> > The rebuttal was helpful and addressed my main concerns. In particular, the additional calibration analysis makes the uncertainty as a reliability measure more convincing. The added results across different domains also strengthen the empirical results.
> > Overall, the rebuttal improved my confidence in paper, and I keep my score unchanged.

---

> > > ### Author Response · Authors · 2026-04-03
> > >
> > > Thank you for your positive feedback. We sincerely appreciate the time and insightful comments you provided in reviewing our paper. **We will incorporate and address all reviewer feedback in the final version.**

---

### Official Review · Reviewer_EtRa · 2026-03-11

**Soundness:** 3
**Presentation:** 3
**Significance:** 3
**Originality:** 2
**Overall Recommendation:** 4
**Confidence:** 3

**Summary:**

This paper proposes UA-Flow, a lightweight extension of flow matching that jointly predicts a heteroscedastic velocity field (mean + diagonal variance) and propagates uncertainty through the deterministic ODE dynamics. The resulting per-sample uncertainty is used in two ways: (i) post-hoc filtering of low-fidelity samples, and (ii) guided sampling via uncertainty-aware classifier guidance (U-CG) and classifier-free guidance (U-CFG). Experiments on CIFAR-10 and ImageNet-128/256 show improved filtering performance over element-wise baselines and moderate generation quality gains from guidance.

**Compliance With Llm Reviewing Policy:**

Affirmed.

**Final Justification:**

Thank you for the detailed rebuttal. My main concerns have been addressed, and I will maintain my score.

**Key Questions For Authors:**

1. In Table 2(c), CFG+U-CG at \lambda=0.5 achieves FID 4.95 vs. CFG-only 5.29. Is this difference statistically significant? What is the variance over multiple evaluation seeds?
2. For U-CG, applying guidance every two steps seems to be a fixed design choice. Was this ablated?

**Strengths And Weaknesses:**

**Strengths**
1. Well-motivated technical design. Modeling uncertainty at the velocity level is a natural and principled choice for flow matching, exploiting its deterministic ODE structure. The conditional reformulation of the NLL loss (Eq. 2) is technically sound, and the derivation in Appendix A is thorough.
2. Computational efficiency. UA-Flow adds only a variance head on top of a pretrained model and requires no retraining from scratch. Table 1 shows it costs 6.075 TFLOPs/image on ImageNet-256, versus 9.731 (BayesDiff) and 33.65 (GenUnc) — a clear practical advantage.
3. Coherent uncertainty maps. Figures 9–10 clearly demonstrate that UA-Flow produces spatially coherent propagated state uncertainty, while BayesDiff degrades to noise-like maps. This qualitative difference is well-explained and supported by Figure 11.

**Weaknesses**
1. Marginal guidance gains. The FID improvements from U-CFG are small in absolute terms: 4.49→4.35 on ImageNet-256 and 5.23→4.82 on ImageNet-128 (Table 3). Given that FID is noisy at this scale, the practical significance is unclear. The paper should include statistical confidence intervals or repeated runs to substantiate these differences.
2. Unfair baseline comparison for filtering. GenUnc is the strongest filter, but the paper dismisses it as "domain-specific" because it uses CLIP embeddings. While this argument is valid, the framing obscures that UA-Flow does not outperform the best available baseline on the core filtering task.

---

> ### Author Rebuttal · Authors · 2026-03-30
>
> We thank the reviewer for the constructive feedback.
>
> ### [W1/Q1] U-CFG FID improvements small; statistical significance.
>
> We rerun U-CFG ($λ_{\max}$=1.5) and CFG (λ=0.75) with 3 seeds on ImageNet-25 6to compute confidence intervals on FID. We find that the confidence intervals are very tight, suggesting that the observed improvement is statistically significant.
>
> |Method|FID↓|Precision↑|Recall↑|
> |-|-|-|-|
> |CFG (λ=0.75)|4.475±0.012|**.7809±.0019**|.4984±.0022|
> |U-CFG ($λ_{\max}$=1.5)|**4.298±0.009**|.7660±.0013|**.5176±.0025**|
>
> Table 2(c) (CFG+U-CG), 3 seeds:
>
> |Method|FID↓|Precision↑|Recall↑|
> |-|-|-|-|
> |CFG (λ=0.5)|5.337±0.042|.7132±.0019|**.5495±.0024**|
> |CFG+U-CG (λ=0.5, w=20)|**4.999±0.047**|**.7281±.0005**|.5393±.0012|
>
> ### [Q2] U-CG every 2 steps — was this ablated? (common response to Reviewer bhpF [W1/Q1] (E) )
>
> We sweep intervals {1, 2, 4} steps across w $\in$ {10, 30, 50} at λ=0 on ImageNet-256. Best FID (18.794) is observed at w=50. Higher-frequency guidance peaks at lower scales since more frequent updates accumulate more total guidance. All configurations improve FID over the unfiltered baseline.
>
> |w|Freq.|FID↓|Prec.↑|Recall↑|
> |-|-|-|-|-|
> |10|1|**20.28**|**.5184**|.6551|
> ||2|21.47|.5105|**.6611**|
> ||4|22.22|.5043|.6600|
> |30|1|**18.91**|**.5284**|.6287|
> ||2|19.45|.5238|.6490|
> ||4|20.82|.5143|**.6604**|
> |50|1|21.42|.5097|.5931|
> ||2|**18.79**|**.5290**|.6359|
> ||4|19.79|.5213|**0.6536**|
>
> ### [W2] GenUnc framing. (common response to Reviewer bhpF [W3] and Reviewer 9dW2 [W2/Q1])
>
> **We find that a straightforward application of our framework in combination with CLIP features can significantly outperform GenUnc.** Since UA-Flow predicts mean and variance at the sampling endpoint, we can draw K=6 latent samples without additional flow-model forward passes, decode them through the VAE, and compute CLIP embedding variance, replicating GenUnc's filtering signal with 2.7× cheaper computational cost.
>
> UA-Flow + CLIP achieves lower FID than GenUnc at every filtering ratio with matched precision, while being 2.7× cheaper. UA-Flow + CLIP requires a single flow-model forward pass (6.1 TFLOPs), plus 5 extra VAE decodes and 6 CLIP encodes for the K=6 samples (6.2 TFLOPS), totaling 12.3 TFLOPS. GenUnc must re-run the full flow ODE multiple times with perturbed weights, costing 33.7 TFLOP. GenUnc retains modestly higher recall, consistent with UA-Flow making a more aggressive precision-recall tradeoff.
>
> More importantly, GenUnc captures only one of UA-Flow's three capabilities (filtering). GenUnc's scalar, post-hoc signal cannot provide gradient-based guidance within the flow ODE. In constrast, UA-Flow's spatially-resolved, model-intrinsic uncertainty uniquely enables U-CG and U-CFG, which are impossible with GenUnc.
>
> ImageNet-256 filtering results:
>
> |Filtering%|UA-Flow+CLIP|||GenUnc|||
> |-|-|-|-|-|-|-|
> ||FID↓|Prec.↑| Rec.↑|FID↓|Prec.↑| Rec.↑|
> |0%|23.45|.5087|.6622|23.23|.5052|.6679|
> |10%|**21.27**|.5156|.6578|22.20|.5146|.6611|
> |20%|**19.40**|.5300|.6512|20.79|.5275|.6577|
> |30%|**18.09**|.5388|.6403|19.37|.5390|.6506|
> |40%|**16.94**|.5565|.6312|18.13|.5544|.6424|
> |50%|**16.22**|.5735|.6154|17.41|.5741|.6332|

---

> > ### Author Rebuttal · Reviewer_EtRa · 2026-04-03
> >
> > All my concerns have been addressed.

---

> > > ### Author Response · Authors · 2026-04-07
> > >
> > > Thank you for the positive assessment. **We will incorporate the concerns you raised into the revised manuscript.**

---

### Official Review · Reviewer_bhpF · 2026-03-13

**Soundness:** 2
**Presentation:** 3
**Significance:** 2
**Originality:** 2
**Overall Recommendation:** 4
**Confidence:** 2

**Summary:**

This paper introduces UA-Flow, which augments a flow matching model with a heteroscedastic variance head and propagates the predicted uncertainty through the deterministic ODE to obtain per-sample uncertainty estimates. These estimates are used in two ways: (i) as a reliability score to filter low-quality samples, and (ii) as a control signal for two guidance variants, uncertainty-aware classifier guidance (U-CG) and adaptive uncertainty-aware classifier-free guidance (U-CFG). Experiments cover CIFAR-10 and ImageNet at 128×128 and 256×256 resolution, with comparisons to BayesDiff, AU, and GenUnc.

**Compliance With Llm Reviewing Policy:**

Affirmed.

**Final Justification:**

Rebuttal adressed some of my concerns, bumped to weak accept.

**Key Questions For Authors:**

- How sensitive are results to the specific design choices (diagonal variance, top-10% aggregation, single-probe covariance, U-CG frequency, choice of f(σ²))? The current ablations are limited; broader ones would help separate principled contributions from tuning.
- Can you provide calibration-oriented validation?
- How important is the bias-correction term in practice? An ablation comparing no correction, the current biased estimator, and perhaps a larger-batch or alternative estimator would clarify this.
- Even a small-scale experiment on a non-image modality (e.g., text, molecular conformations) would materially support the modality-agnostic framing.

**Limitations:**

Yes

**Strengths And Weaknesses:**

## Strengths

- The overall recipe of variance head → ODE propagation → filtering/guidance, is easy to state and straightforward to implement on top of any flow matching backbone.
- UA-Flow is considerably cheaper than the ensemble or sampling-based uncertainty baselines it compares against, while still producing informative uncertainty maps.
- The paper evaluates filtering, U-CG, U-CFG, and cost, and includes appendix experiments supporting the correlation assumption underlying U-CFG.
- The appendix acknowledges the Jensen bias incurred by squaring the self-normalized importance estimator in the loss, which I appreciated.

## Weaknesses
- The method stacks several approximations and heuristics: a biased importance-weighted surrogate loss, first-order linearization for variance propagation (despite second-order Heun sampling), diagonal variance, a hand-chosen pseudo-likelihood f(σ²), top-10% aggregation, sparse uncertainty updates, and single-probe covariance estimation. Each is individually defensible, but their combined effect is never disentangled. The result reads more as a pragmatic recipe than a validated probabilistic formulation. This is the main thing I think should be studied better.
- Maybe I don't understand it well enough, but the primary evidence for the quality of the uncertainty estimates is that filtering high-uncertainty samples improves FID/precision. This supports usefulness as a ranking heuristic, but says little about whether the estimates are calibrated or monotonically related to a meaningful notion of sample error. For a paper that frames its contribution in terms of uncertainty quantification, calibration analysis would substantially strengthen the claims.
- GenUnc achieves lower post-filtering FID than UA-Flow. The paper argues GenUnc is domain-specific (CLIP-based), which is fair, but it does weaken the practical claim that UA-Flow provides the best reliability signal for images.
- The paper repeatedly invokes "modality-agnostic" applicability and "safety-critical settings," yet all experiments are image generation. Without even a small non-image experiment, this framing is aspirational rather than substantiated and should be toned down.

---

> ### Author Rebuttal · Authors · 2026-03-30
>
> We thank the reviewer for the thorough evaluation.
>
> ### [W1/Q1] Combined effect of approximations; limited ablations.
>
> We address each concern with new ablations on ImageNet-256. The ablated configurations target memory for uncertainty prediction (A), uncertainty reduction for filtering (B), runtime from update frequency (C), Jacobian estimation (D), and U-CG design choices (E & F). **The ablations demonstrate that each choice is robust, and that together they make the method tractable without sacrificing the core uncertainty signal.**
>
> **(A) Diagonal variance.** A full covariance over the high-dimensional output would require storing its squared dimensionality. Diagonal approximation is standard in high-dimensional UQ (e.g.[1]).
>
> [1] A. Kendall & Y. Gal. "What Uncertainties Do We Need in Bayesian Deep Learning for Computer Vision?" NeurIPS 2017.
>
> **(B) Top-X% aggregation.** We sweep X from 5& to 50% at 50% filtering. FID and precision improvement are consistent across the entire range although exact values vary across aggregation settings:
>
> |Aggregation|FID↓|Prec↑|Rec↑|
> |-|-|-|-|
> |5%|21.24|.5210|.6505|
> |10%|**21.02**|**.5216**|.6519|
> |25%|21.37|.5192|.6525|
> |50%|22.23|.5164|**.6527**|
>
> **(C) Sparse update frequency.** We ablate the variance-head update interval in {2, 4, 8} steps during filtering. At 50% removal, all three are closely matched. The default (4) halves the overhead of every-2 with no degradation:
>
> |Interval|FID↓|Prec↑|Rec↑|
> |-|-|-|-|
> |2|21.45|.5189|.6511|
> |4|21.06|.5177|.6454|
> |8|**21.00**|.5180|.6485|
>
> **(D) Single-probe covariance.** Ablated in Appendix G.3.2. Single-probe estimation closely matches multi-probe estimates.
>
> **(E) U-CG frequency.** Please see our response to Reviewer EtRa [Q2]. U-CG with different frquencies improve over the unfiltered baseline.
>
> **(F) Pseudo-likelihood $f(σ^2)$ variants.** Three formulations at w=50, λ=0. All three improve over the baseline. The default ($-(\mathrm{mean}(σ^2))^2$) is within 0.15 FID of the best, showing variate choice of pseudo-likelihood.
>
> |$f(\sigma^2)$|FID↓|Prec↑|Rec↑|
> |-|-|-|-|
> |$-(\mathrm{mean}(σ^2))^2$|18.80|.5290|.6359|
> |$-(\mathrm{mean}(σ))^2$|**18.65**|**.5318**|**.6470**|
> |$-\mathrm{mean}((σ^4)^2)$|20.22|.5229|.6227|
>
> ### [W2/Q2] Calibration analysis.
>
> We design an indirect calibration protocol because generative models lack ground-truth but target to sample from the data distribution: we partially noise real ImageNet validation images to $z_t$ and compare the predicted endpoint $\hat{z}_1$ against the true latent $z_1$. We evaluate state uncertainty using PIT ECE and Brier score over all tensor components on 500 images. We compare UA-Flow against BayesDiff and AU; GenUnc is excluded because it produces only a scalar uncertainty. AU is only measured at t=.9 since it quantifies uncertainty near the endpoint.
>
> **UA-Flow consistently outperforms BayesDiff and AU both before and after calibration**. The scores improves monotonically with t since low-t trajectories resemble random generation rather than reconstruction.
>
> **Before calib. (PIT ECE↓ / Brier↓)**
>
> |t|UA-Flow|BayesDiff|AU|
> |-|-|-|-|
> |.5|.1328/.4771|.2381/.9025|—|
> |.7|.0992/.3281|.2381/.9025|—|
> |.9|**.0277/.1009**|.2386/.9025|.0485/.1782|
>
> **After calib. (PIT ECE↓ / Brier↓)**
>
> |t|UA-Flow|BayesDiff|AU|
> |-|-|-|-|
> |.5|.0066/.0515|.0105/.0558|—|
> |.7|.0038/.0492|.0073/.0546|—|
> |.9|**.0008/.0481**|.0038/0.0505|.0012/.0496|
>
> ### [Q3] Bias-correction ablation.
>
> We train a model without the bias-correction term and evaluate all three downstream tasks (filtering/U-CG/U-CFG) on ImageNet-256. Replacing the correction with a stop-gradient velocity prediction causes training instability, so only the two tractable conditions are compared.
>
> **Filtering and U-CG** perform comparably in both conditions. Relative filtering FID reductions are comparable (Δ2.22vsΔ2.36). FID improvements and precision-recall tradeoff from both tasks preserved without the correction.
>
> However, results of **U-CFG** are different. Without correction, U-CFG ($λ_{\max}$=1.25) cannot surpass fixed CFG (λ=0.75) because the variance head overestimates uncertainty at early sampling stages, yielding excessive guidance early unlike U-CFG with correction: https://i.postimg.cc/d0q7c6QZ/correction.png
>
> |Task|FID↓|Prec↑|Rec↑|
> |-|-|-|-|
> |Filtering (50%)|20.66|.5158|.6462|
> |U-CG (w=50)|18.44|.5296|.6389|
> |Best U-CFG|4.666±0.052|.7578±.0040|.5063±.0029|
> |Best CFG|**4.529±0.036**|.7821±.0066|.4920±.0037|
>
> ### [W3] GenUnc achieves lower post-filtering FID.
>
> Please see our response to Reviewer EtRa [W2]. In brief: we find that a simple modification to our method allows computing CLIP feature variance and use this to demonstrate superior post-filtering FID with lower computations than GenUnc.
>
> ### [W4/Q4] Non-image experiments.
>
> Please see our response to Reviewer 9dW2 [W4]. In brief: Through experiments on a 2D checkerboard and sine wave time-series, we show that UA-Flow's uncertainty quantification generalizes to non-image domains.

---

> > ### Author Rebuttal · Reviewer_bhpF · 2026-04-06
> >
> > The rebuttal addresses several of my original concerns and I appreciate the amount of additional experimental evidence provided. In particular, the new ablations on aggregation, update frequency, pseudo-likelihood choice, and the bias-correction term help clarify that the method is reasonably robust to a number of implementation choices. I also appreciate that the authors added a calibration-oriented evaluation, since this was one of my main requests.
> >
> > That said, my central concern is only partially resolved. The work still appears to rely on a stack of approximations and heuristics whose combined probabilistic meaning remains somewhat unclear. The new ablations are useful, but they mostly support robustness of the recipe rather than a principled uncertainty-quantification formulation. Similarly, the calibration experiment strengthens the paper, but it is indirect and reconstruction-based rather than a fully convincing validation of uncertainty quality for free-running generative sampling.
> >
> > The additional non-image experiments are helpful for softening my concern about the “modality-agnostic” framing, but from the rebuttal they still appear small in scale relative to the main image results, so I think the broader framing should remain modest. Overall, I view the rebuttal as a meaningful improvement and as partially resolving my questions, but not enough to fully change my assessment of the paper’s core strengths and weaknesses.

---

> > > ### Author Response · Authors · 2026-04-07
> > >
> > > We sincerely thank the reviewer for the thoughtful evaluation and for recognizing the value of our additional experiments. We address the remaining concerns below.
> > >
> > > ### 1. Combined probabilistic meaning of approximations
> > >
> > > We would like to clarify the probabilistic interpretation and justifications behind our approximations and design choices. We emphasize that these choices reflect a deliberate **balance between computational practicality and theoretical completeness.**
> > >
> > > **First-order linearization for variance propagation** is introduced to reduce computational overhead. For example, in Heun2, the intermediate velocities $u_1:=u_t(x_t)$ and $u_2:=u_{t+dt/2}(x_t+u_1dt/2)$ are used for the next-step integration. Full variance propagation would require three covariance terms: $\mathrm{Cov}(x_t,u_1)$, $\mathrm{Cov}(x_t,u_2)$, and $\mathrm{Cov}(u_1,u_2)$, each demanding a separate JVP computation for $u_1$ and $u_2$, which at least doubles the computational cost. We will include the full derivation of variance propagation under Heun2 in the revised manuscript.
> > >
> > > **Single-probe covariance estimation.** The consistent performance of single-probe estimation in filtering may indicate near-diagonality of the Jacobian. Single-probe estimation becomes identical to the target approximation (Eq. (6)) when the Jacobian is exactly diagonal. We observe consistent performance which suggests that the Jacobians encountered in our filtering tasks are indeed well approximated as diagonal.
> > >
> > > **Top-X% uncertainty aggregation** is a sample estimator of the Conditional Value-at-Risk (CVaR) at level α=X/100 [1]. For element uncertainty Z in each sample, $\mathrm{CVaR}_α(Z)=E[Z|Z≥Q_{1-α}(Z)]$, where $Q_{1-α}$ is the 1-α quantile, empirically the average over the top-α fraction. CVaR is the standard tail-risk objective.
> > >
> > > **Pseudo-likelihood f(σ²).** U-CG follows directly from the standard guidance derivation. Treating the predicted uncertainty σ² as an observation to be small, the guided velocity is $$u_t(x|σ²) = a_tx+b_t∇\log p_t(x|σ²)=u_t(x)+b_tw∇\log p_t(σ²|x),$$ where w is the guidance scale and $u_t(x)=a_tx+b_t∇\log p_t(x)$. The pseudo-likelihood $\log p_t(σ²|x)$ must be a monotonically decreasing function of σ². Therefore, the observed robustness across $f(σ²(x))$ in the ablation in the rebuttal is expected.
> > >
> > > **Others.** Diagonal variance and bias-correction term are tractability-driven approximations, as the exact representations are infeasible under the memory and compute constraints of high-dimensional generative modeling as described in the main paper and the rebuttal. Additionally, while sparse uncertainty updates are introduced primarily to reduce computational overhead, the ablation study confirms that this approximation incurs no measurable loss in filtering performance.
> > >
> > > * [1] R. Rockafellar et al. "Optimization of conditional value-at-risk." Journal of risk, 2000.
> > >
> > > ### 2. Calibration experiment is indirect and reconstruction-based
> > >
> > > We appreciate the reviewer's desire for a direct calibration validation. However, **to the best of our knowledge, there is no standard calibration protocol for flow-based generative models and none of our baselines (BayesDiff, AU, GenUnc) provide any form of calibration analysis.**: when generating novel samples from pure noise, there is no ground-truth target to compare against, neither the "true" clean sample nor the "true" velocity field exists for a freely generated trajectory. This is an inherent property of generative models. Our reconstruction-based protocol is the first calibration analysis for uncertainty estimation in flow-based generative models.
> > >
> > > ### 3. Non-image experiments remain small-scale
> > >
> > > Please refer to our reply to Reviewer 9dW2 for the full results and protocol. We share this concern and have added a **PushT robot manipulation experiment**. PushT is a sequential decision-making task in which a robot policy generates action chunks to push a T-shaped block to a target pose, with **25,650 training samples of paired image observations and robot states**. Applying UA-Flow uncertainty-based filtering yields clear performance improvements across all metrics.
> > >
> > > Independent of this additional evidence, we agree that the broader framing should be described carefully, and we will revise the language from **modality-agnostic** to **extendable to non-image domains** accordingly in the revised version.

---

### Decision · Program_Chairs · 2026-04-30

**Decision:**

Reject

**Comment:**

This submission develops UA-flow a method that extends flow matching with a variance head for uncertainty quantification. UA-flow adds a diagonal covariance head to the network that predicts the velocity, which induces a Gaussian distribution over implied end point estimates. This can then be used to define uncertainty-aware guidance strategies (both classifier-based and classifier-free) in which the uncertainty of the end point estimate serves to modulate the guidance strength. Experiments on CIFAR-10 and ImageNet-128/256 show that UA-Flow's uncertainty is better correlated with sample fidelity than baselines (BayesDiff, AU, GenUnc), and that uncertainty-guided sampling improves FID and precision over standard CFG.

While reviewers are leaning accept on this submission, we have a very competitive pool this year, which regrettably means that this submission is borderline in terms of scores. Noted strengths are the simplicity and computational efficiency of the implementation. Weakneses are that the approach stacks several levels of approximation and that empirical results, while encouraging, are a bit marginal. The authors provided additional ablations and calibrations as part of the author response, some additional non-image experiments on toy datasets and committed to revising some aspects of framing. One reviewer raised their score. One reviewer maintained their weak reject score, noting overall concerns about the effectiveness of the method, particularly in high-dimensional settings (which were not entirely addressed by additional experiments on low-dimensional data).

Overall this was a submission that could have gone either way. Owing to the fact that we have a lot of strong paper this year, the submission unfortunately had to be cut during calibration. We wish the authors the best of luck with a resubmission to a future venue.